# Disrupted glucocorticoid receptor cell signalling causes a ciliogenesis defect in the fetal mouse renal tubule

Kelly L Short [iD][1,3], Jianshen Lao[1,3], Rachel Lam[2], Julie L M Moreau[2], Judy Ng[1], Mehran Piran[2], Alexander N Combes[2], Denny L Cottle [iD][2] & Timothy J Cole [iD][1✉]

## Abstract

Primary cilia are cell signalling and environment sensing organelles and have important roles during embryogenesis and homeostasis. We demonstrate glucocorticoid signalling is essential for normal cilia formation in mouse and human renal tubules. RNA sequencing of E18.5 kidneys from glucocorticoid receptor (GR) null mice identified significant reductions in key ciliogenesis-related genes including Ccp110, Cep97, Cep290 and Kif3a. Confocal microscopy reveals abnormal, stunted cilia on proximal tubules, podocytes, and collecting duct cells in mice with global or conditional deletion of GR. In contrast, activation of GR signalling with dexamethasone in human kidney organoids or mouse IMCD3 cells increases cilia length, an effect blocked by the GR antagonist RU486. Analysis of GR-null kidney extracts demonstrates reduced levels of pERK and SUFU identifying potential cell pathway crosstalk with GR signalling that coordinately regulate ciliogenesis in the renal tubule. Finally, dexamethasone reduces Aurora kinase A levels, a factor driving cilia disassembly and implicated in the pathogenesis of polycystic kidney disease.

**Keywords** Ciliogenesis; Ciliopathies; Glucocorticoids; Glucocorticoid Receptor Signaling; Kidney Development
**Subject Categories** Cell Adhesion, Polarity & Cytoskeleton; Urogenital System

## Introduction

Primary cilia are microtubule-based organelles found protruding from the surface of most mammalian cells (Satir et al, 2010; Zhao et al, 2023). The primary cilium mediates diverse developmental signalling pathways and senses extracellular stimuli to maintain tissue homeostasis. In the kidney, primary cilia are located on epithelial cells of tubules projecting into the lumen where they function as fluid sensors (Pluznick and Caplan, 2015). Primary cilia are 1–10 μm long and 0.2–0.3 μm wide and comprise four main parts: the axoneme, basal body, ciliary membrane and the transition zone (Zhao et al, 2023). The axoneme contains nine microtubules arranged as a 9 + 0 ring structure attached to the cell membrane (Zhao et al, 2023). Defects in primary cilia function or structure cause a group of human diseases called ciliopathies including Polycystic Kidney Disease (PKD) (McConnachie et al, 2021). A clinical case report of a preterm infant with advanced PKD has documented improved PKD outcomes following treatment with the synthetic glucocorticoid steroid dexamethasone (Katz et al, 1979). In addition, a study in human adipose stem cells, showed that dexamethasone treatment in vitro promoted cilia growth and elongation, and dexamethasone has also been shown to restore primary cilia gene expression in cancer cells (Forcioli-Conti et al, 2015). Additionally, a recent study demonstrated that aldosterone acting via the mineralocorticoid receptor in renal collecting ducts could regulate cilia length (Komarynets et al, 2020), and in a separate study, glucocorticoids regulate cilia gene expression in hippocampal neurones (Mifsud et al, 2021). Given these observations we have investigated the impact of glucocorticoid signalling on ciliogenesis in glucocorticoid receptor null mice, glucocorticoid treated human organoid cultures and renal cell lines.

Glucocorticoid steroid hormones play essential roles for maturation and growth of many fetal organs including the lung and heart, yet kidney-specific roles are not well characterised (Fowden and Forhead, 2015; Whirledge and DeFranco, 2018). Glucocorticoids activate the intracellular glucocorticoid receptor (GR) that acts primarily as a nuclear transcriptional regulator (Ackermann et al, 2010; Cole and Young, 2017; Quax et al, 2013). Global GR-null mice die shortly after birth due to respiratory failure (Bird et al, 2014; Bird et al, 2007; Cole et al, 1995) and several studies have characterised mouse models of conditional GR deletion in kidney epithelium. Deletion in the distal nephron using a *Ksp*-cre driver resulted in a mild increase in blood pressure in adult mice with normal renal histology (Goodwin et al, 2010). A more extensive renal tubular deletion of *GR* using an inducible *Pax8*-cre driver altered the expression of several epithelial sodium transporters including the thiazide-sensitive Na⁺/Cl⁻ cotransporter (*Ncc*), the Na-K-Cl cotransporter-2 (*Nkcc2*) (Canonica et al, 2019). Furthermore, adult mice heterozygous for a global GR-null allele developed salt-sensitive hypertension and reduced mRNA levels of *Ncc* when on a high salt diet (Ivy et al, 2018). Additionally, rodents exposed to synthetic glucocorticoids have reduced birth weight and

[1]Department of Biochemistry and Molecular Biology, Monash University, Melbourne, VIC 3800, Australia. [2]Department of Anatomy and Developmental Biology, Monash University, Melbourne, VIC 3800, Australia. [3]These authors contributed equally: Kelly L Short, Jianshen Lao. ✉E-mail: tim.cole@monash.edu

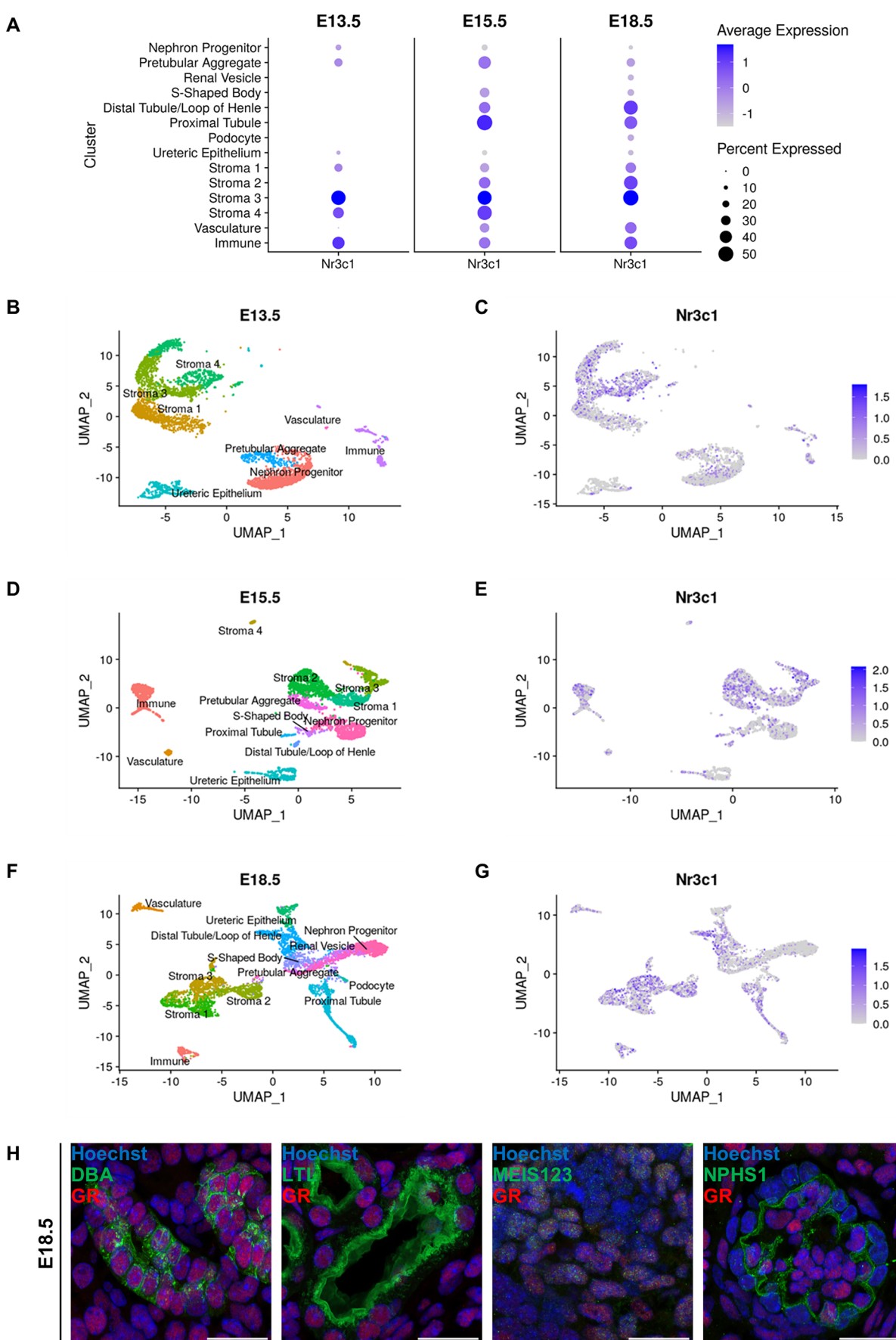

**Figure 1.  Glucocorticoid receptor gene expression in the fetal kidney during embryonic development.**

(**A**) Average expression of GR across the cell clusters at E13.5, E15.5 and E18.5. (**B**) tSNE plot showing cell clusters within the kidney at E13.5. (**C**) tSNE plot showing the expression of GR, purple within the cell clusters at E13.5. (**D**) tSNE plot showing cell clusters within the kidney at E15.5. (**E**) tSNE plot showing the expression of GR, purple within the cell clusters at E15.5. (**F**) tSNE plot showing cell clusters within the kidney at E18.5. (**G**) tSNE plot showing the expression of GR, purple within the cell clusters at E18.5. (**H**) Immunofluorescence of glucocorticoid receptor (GR) in the fetal kidney during embryonic development at E18.5. Sections were stained Hoechst (blue, nucleus), Dolichos Biflorus Agglutinin (DBA) (green, collecting duct), Lotus Tetragonolobus Lectin (LTL) (green, proximal tubule), nephrin (NPHS1) (green, podocyte), MEIS123 (green, stroma) and GR (red, GR). Slides were imaged with a Zeiss LSM 980 confocal microscope (×63 objective, 2× digital zoom), scale bar represents 20 μm. All images are representative of $n = 4$ animals per experimental group. Source data are available online for this figure.

increased risk of high blood pressure in adulthood (Seckl, 2004). It has been shown that both glucocorticoid excess and deficiency can lead to salt-sensitive hypertension (Bailey et al, 2009; Ivy et al, 2018). We show here that GR-medicated glucocorticoid signalling regulates primary ciliogenesis in the renal tubule. Our results demonstrate a capacity for GR agonists and antagonists to modulate cilia formation and function.

## Results

### Localisation of the GR in the developing kidney during embryogenesis

While loss of *GR* is linked to hypertension and changes in solute transporter gene expression, the cellular expression profile and role of *GR* during kidney development remains unclear. The expression of *GR* was investigated using single-cell RNA sequencing datasets from the developing mouse kidney at embryonic day (E)13.5, E15.5, and E18.5 (Combes et al, 2019). Expression of the *GR* gene (official symbol *Nr3c1*) was observed within early nephron, ureteric epithelium, stromal and immune cell clusters at E13.5, with highest expression within stromal clusters 3 and 4 which are marked by collagen type III alpha 1 (*Cola1*), Decorin (*Dcn*), Delta like non-canonical Notch ligand 1 (*Dlk1*) and Periostin osteoblast specific factor (*Postn*) (Figs. 1A–C and EV1A). *GR* expression was detected in the proximal tubule, distal tubule/loop of Henle, ureteric epithelium/collecting duct, immune cells and vasculature at E15.5 and E18.5 (Figs. 1D–G and EV1B,C). As such, GR signalling has the potential to impact most stromal and epithelial cell types in the developing kidney.

To compare GR localisation at the protein level at different stages of fetal kidney development, we performed immunofluorescence using a GR antibody with markers of the proximal tubules (lotus tetra-gonolobus lectin, LTL), collecting duct (dolichos biflorus agglutinin, DBA), podocytes (nephrin, NPHS1) and stroma (MEIS123). At E14.5, GR was strongly localised to the developing collecting duct, limited in the proximal tubule (Davidson, 2009; Short and Smyth, 2016) compartments consistent with the single cell data from E13.5 to E15.5 (Fig. 1A; Appendix Fig. S1). At E16.5 and E18.5 GR is widely localised in the kidney with positive staining in the proximal tubules, collecting ducts, podocytes and stroma (Fig. 1H; Appendix Figs. S1 and S2), suggesting GR signalling having important roles in these structures during kidney development.

### GR-null mice at E18.5 have an altered renal transcriptome

The role of GR-mediated signalling in the developing fetal kidney was analysed in GR-null mice at E18.5. Loss of GR was confirmed

by immunohistochemistry and western blot analysis (Cole et al, 1995) (Fig. EV2). Histological analysis by periodic acid Schiff staining of control and GR-null littermate kidney showed no major abnormalities in renal size or structures at E18.5 (Appendix Fig. S3). As the GR is a steroid ligand-activated transcriptional regulator we then analysed the effect of loss of GR expression on the fetal kidney transcriptome at E18.5 by RNA sequencing. Total RNA was extracted from control ($n = 4$) and GR-null ($n = 3$) kidneys and analysed by NGS RNA-seq. Global changes in mRNA levels were displayed as a heatmap for genes with a fold change greater than 1 and a false discovery rate (FDR) of less than 0.05 (Fig. 2A) and principal component plot showing separation of control and GR-null kidney at E18.5 (Fig. 2B). Loss of GR expression resulted in 2473 differentially expressed genes (FDR < 0.05), 288 genes with absolute LogFC >1 & FDR < 0.05 (Fig. 2C; Table EV1), which identified 16 upregulated and 25 downregulated ciliary genes (FDR < 0.05) (Fig. 2D). To confirm gene expression changes detected by RNA sequencing, qRT-PCR was performed to quantify mRNA levels of twenty selected differently expressed protein coding genes. There was a significant decrease in GR-null fetal kidney mRNA levels at E18.5 in downregulated genes compared to controls (Fig. 2E–G). scRNA-seq data was used to identify the localisation of the fifteen selected differentially expressed genes within the E18.5 embryonic kidney. *Igf2* was expressed across the kidney while other genes had a more restricted expression (Fig. 2H). Gene set enrichment analysis identified increased signatures of AKT (Nishimura et al, 2021), ERK/MAPK (Kuonen et al, 2019) and mTOR (Lai and Jiang, 2020) pathways which have been linked to regulate or be regulated by primary ciliogenesis and function (Fig. EV3A).

We next employed a deconvolution approach, where scRNA-seq data was used to estimate changes in cell type signatures or cell proportions within the bulk RNA-seq dataset (Fig. EV3B,C). Small but significant reductions in the estimated proportions of early proximal tubule and medullary stromal populations (which express higher levels of *GR*), were identified in *GR*-null kidneys, suggesting subtle changes in the development of *GR*-expressing cell types (Fig. EV3B,C).

### GR-null mice have a transcriptional signature of dysregulated ciliogenesis and stunted primary cilia on renal proximal tubule cells

Transcriptome sequencing of whole GR-null fetal kidney at E18.5 showed a reduction in a number of key ciliogenesis genes, including *Ccp110, Cep97, Cep290, Kif3a* and *Rpgr* which indicated a potential deficit in primary ciliogenesis. qRT-PCR analysis confirmed downregulation of these cilia-associated genes in

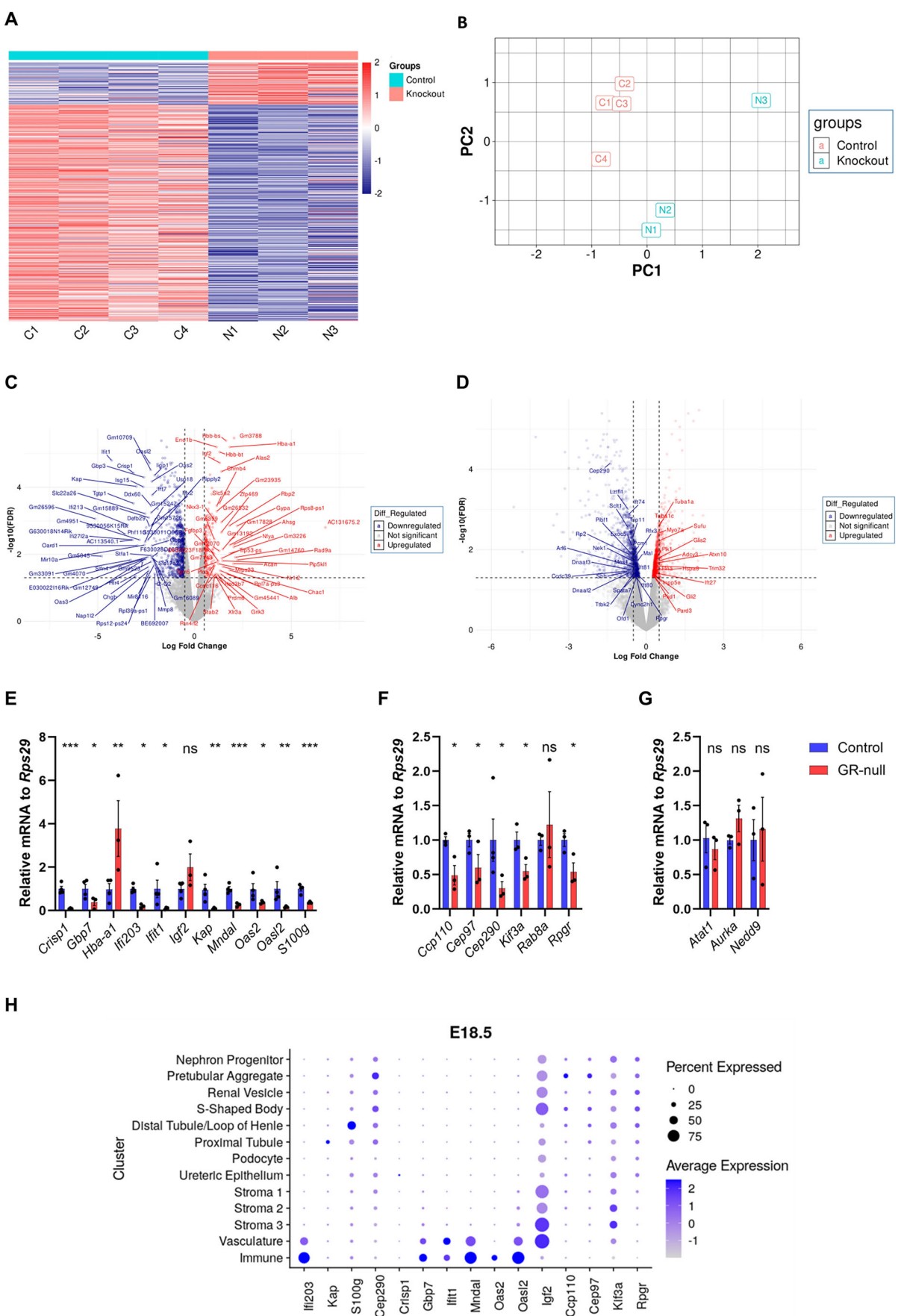

◀ **Figure 2. Transcriptome analysis of E18.5 fetal kidney RNA from GR-null and control mice.**

(A) NGS RNA-seq was performed on total RNA isolated from control and GR-null mouse kidneys at E18.5. Heatmaps generated from differentially expressed genes (LogFC >1 and FDR < 0.05) log2-CPM values. The number of suppressed genes is four times more than augmented genes in GR-null compared to control. Data from control (*n* = 4) and GR-null (*n* = 3) animals per experimental group. (B) Principal component plot showing separation of control (red) and GR-null (blue) kidney at E18.5. Data from control (*n* = 4) and GR-null (*n* = 3) animals per experimental group. (C) Volcano plot of top differentially expressed genes with a significant FDR < 0.05 and log-fold change >1.5 and < −2. Red dots represent gene mRNA levels that are significantly increased, and blue dots represent genes with significantly decreased. Dashed lines show the LogFC 0.5 and FDR 0.05 cut-offs. Data from control (*n* = 4) and GR-null (*n* = 3) animals per experimental group. (D) Volcano plot of differentially expressed ciliary genes with a significant FDR < 0.05. Red dots represent gene mRNA levels that are significantly increased, and blue dots represent genes with significantly decreased. Dashed lines show the LogFC 0.5 and FDR 0.05 cut-offs. Data from control (*n* = 4) and GR-null (*n* = 3) animals per experimental group. (E) mRNA levels of eleven target genes identified from NGS RNA-seq in fetal GR-null mouse kidney at E18.5. *Crisp1* (fold −9.6, *P* = 0.00030), *Gbp7* (fold −3.7, *P* = 0.012), *Ifi203* (fold −5.6, *P* = 0.00073), *Ifit1* (fold −8.6, *P* = 0.024), *Kap* (fold −9.2, *P* = 0.0024), *Mndal* (fold −3.8, *P* = 0.00046), *Oas2* (fold −2.6, *P* = 0.016), *Oasl2* (fold −4.6, *P* = 0.0062) and *S100g* (fold −2.7, *P* = 0.00094) were downregulated. Consistent with RNA sequencing *Hba-A1* (fold 3.9, *P* = 0.049) and *Igf2* (fold 1.7, *P* = 0.071) were upregulated. The mRNA levels are expressed relative to mRNA levels of the housekeeping gene *Rps29*. All data presented as mean ± SEM, significant differences were analysed by unpaired T tests indicated by *\**$P$ ≤ 0.05, \*\**P* ≤ 0.01, \*\*\**P* ≤ 0.001, \*\*\*\**P* ≤ 0.0001, ns=not significant, between control and GR-null samples, *n* = 3 animals per experimental group for each gene. (F) mRNA levels of six primary cilia structural genes in fetal GR-null mouse kidney at E18.5. *Ccp110* (fold −2.2, *P* = 0.011), *Cep97* (fold −1.8, *P* = 0.048), *Cep290* (fold −2.9, *P* = 0.012), *Kif3a* (fold −1.8, *P* = 0.026), *Rab8a* (fold 1.1, *P* = −0.80) and *Rpgr* (fold −1.9, *P* = 0.026). The mRNA levels are expressed relative to mRNA levels of the housekeeping gene *Rps29*. All data presented as mean ± SEM, significant differences were analysed by unpaired T tests indicated by *\**$P$ ≤ 0.05, \*\**P* ≤ 0.01, \*\*\**P* ≤ 0.001, \*\*\*\**P* ≤ 0.0001, ns=not significant, between control (*n* = 3–4) and GR-null (*n* = 3) animals for each gene. (G) mRNA level of three primary cilia regulatory genes in fetal GR-null mouse kidney at E18.5. *Atat1* (fold −1.19, *P* = 0.57), *Aurka* (fold 1.32, *P* = 0.19) and *Nedd9* (fold 1.16, *P* = 0.79). The mRNA levels are expressed relative to mRNA levels of the housekeeping gene *Rps29*. All data presented as mean ± SEM, significant differences were analysed by unpaired T tests indicated by *\**$P$ ≤ 0.05, \*\**P* ≤ 0.01, \*\*\**P* ≤ 0.001, \*\*\*\**P* ≤ 0.0001, ns=not significant, between control and GR-null, *n* = 3 animals for each gene. (H) Expression of target genes within different cell types of the kidney at E18.5. Scale represents average expression, generated from single cell dataset. Source data are available online for this figure.

GR-null mouse kidney with significant reductions in mRNA levels of *Ccp110* (fold −2.17, *P* = 0.011), *Cep97* (fold −1.79, *P* = 0.047), *Cep290* (−2.9-fold, *P* = 0.012), *Kif3a* (fold −1.82, *P* = 0.026) and *Rpgr* (fold −1.87, *P* = 0.025) (Fig. 2F), but no changes *Rab8a* (fold 0.921, *P* = 0.805) or in ciliary genes known to regulate primary cilia; *Atat1* (fold −1.19, *P* = 0.573), *Aurka* (fold 0.778, *P* = 0.148) and *Nedd9* (fold 0.879, *P* = 0.802) (Fig. 2G). To investigate further, the primary cilium was visualised in the proximal tubule of the GR-null fetal kidney by immunofluorescence (Fig. 3A). Acetylated tubulin and ARL13B were stained to confirm co-localisation of the cilia axoneme in GR-null fetal kidney (Fig. EV4A). Kidneys were stained with acetylated tubulin to mark microtubules of the cilia axoneme, LTL to mark proximal tubules and pericentrin to mark the basal body. Compared to control mice, primary cilia located on kidney proximal tubule cells were stunted and abnormal in the GR-null mouse (Fig. 3A). Primary cilia length on GR-null kidney proximal tubule cells (5.10 ± 0.11 μm) was significantly decreased compared to control mice (6.20 ± 0.15 μm) (Fig. 3B). Additionally, the percentage of primary cilia longer than 5 μm was significantly lower on GR-null proximal tubule cells (38.47 ± 4.21%) when compared to controls (56.38 ± 4.45%) (Fig. 3C). There was no significant difference in the percentage of ciliated proximal tubule cells between GR-null mice and controls (Fig. 3D). The presence of abnormally shaped cilia in the renal tubule of GR-null mice, compared to controls, was further confirmed with scanning electron microscopy images that showed abnormal primary cilia morphology with many cilia displaying bulging areas and an abnormal shape (Fig. 3E). Primary cilia structure in other renal tubule segments such as the collecting ducts could not be assessed at E18.5 because the intralumenal space was too constricted at this stage of development (Fig. EV4B). Finally, in contrast to the fetal lung of GR-null mice, immunohistochemistry staining for Ki67, a marker of cell proliferation, showed no change in cell proliferation in the fetal kidney at E18.5, both in the mesenchymal compartment and in the proximal tubule (Bird et al, 2014; Bird et al, 2007) (Fig. EV5A–C).

## GRcdKO mice have stunted cilia on renal collecting duct cells at PND11

The global GR-null mouse dies at birth due to respiratory failure, therefore further analysis of a cilia phenotype postnatally in the renal tubule was not possible. To allow further analysis postnatally, conditional deletion of the GR in collecting ducts of the kidney was achieved using a HoxB7-Cre recombinase transgenic mouse crossed to a GR-loxP/loxP mouse strain. The effect of GR loss on collecting duct primary cilia morphology was analysed using a conditional collecting duct GR deletion at postnatal day 11 (Fig. 4A). Postnatal day 11 was selected for primary analysis because mouse models of PKD showed prominent cysts by 2 weeks in age (Rachel et al, 2015). Kidneys were stained with acetylated tubulin to mark microtubules, DBA to stain collecting ducts, and pericentrin to visualise the basal body of cilia. Primary cilia length on GRcdKO kidney collecting ducts cells were significantly decreased (3.30 ± 0.06 μm) when compared to control (3.63 ± 0.05 μm) (Fig. 4B). There was no significant difference in the percentage of primary cilia longer than 5 μm in GRcdKO kidney collecting ducts (18.09 ± 1.24%) when compared to controls (22.21 ± 8.48%) (Fig. 4C). In addition, there was no significant difference in the percentage of ciliated collecting duct cells between GRcdKO kidney collecting ducts (66.38 ± 4.56%) and control (65.88 ± 3.58%) (Fig. 4D).

## GR-null mice have stunted primary cilia on kidney podocytes at E18.5

Immunofluorescence was used to assess the morphology of primary cilia on GR-null kidney podocytes. Kidneys were stained with acetylated tubulin and nephrin which stains the podocytes. Primary cilia on GR-null kidney podocytes also appeared stunted when compared to controls (Fig. 5A). Primary cilia length on GR-null kidney podocytes was significantly decreased (2.43 ± 0.09 μm) when compared to controls (2.73 ± 0.10 μm) (Fig. 5B). There was no significant difference in the percentage of primary cilia longer than

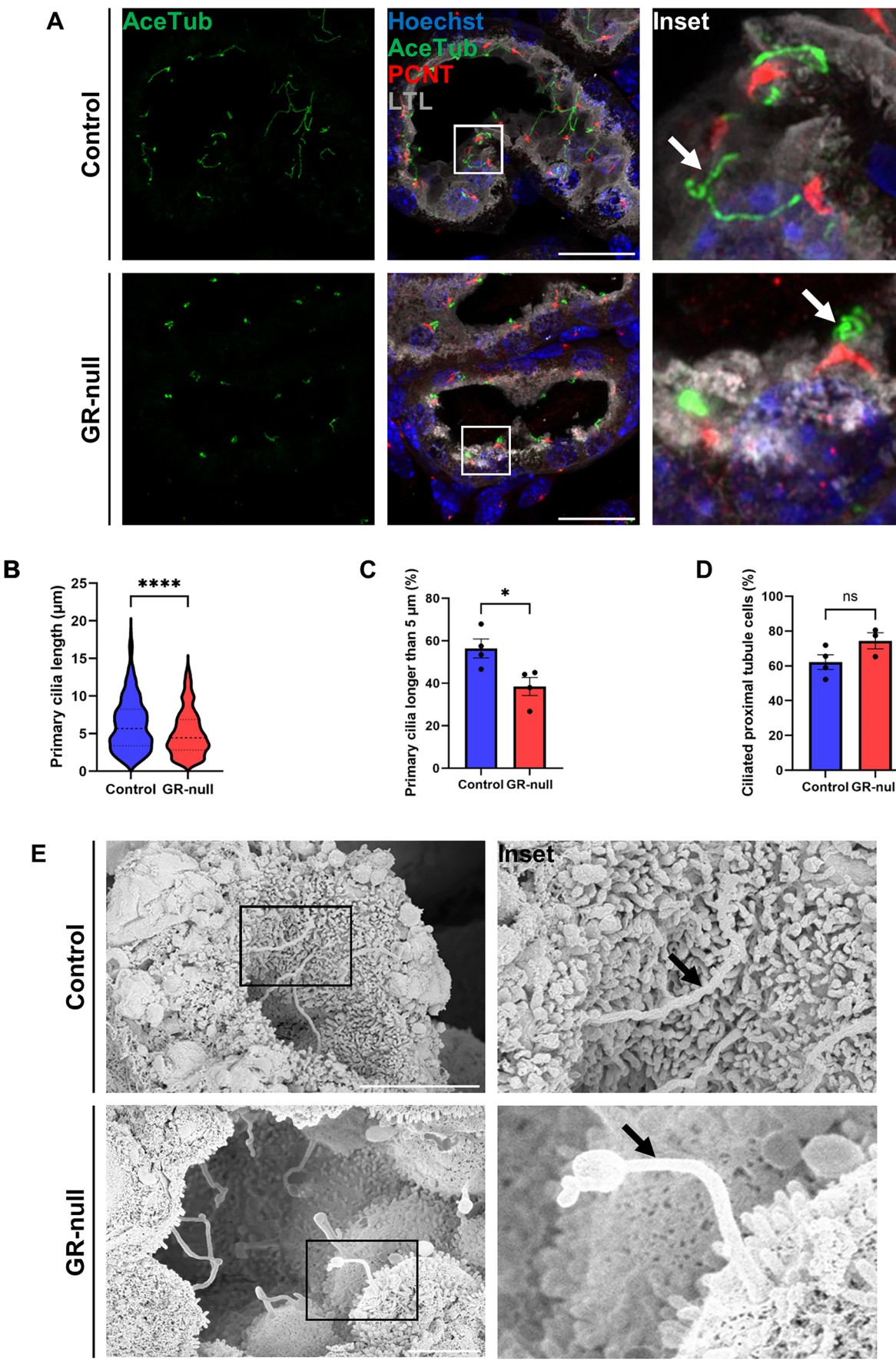

**Figure 3. Glucocorticoid regulation of primary cilia length on GR-null fetal kidney proximal tubule cells at E18.5.**

(A) Immunofluorescence of primary cilia morphology in control and GR-null proximal tubules at E18.5. Sections were stained with Hoechst (blue, nucleus), acetylated tubulin (AceTub) (green, microtubules), pericentrin (PCNT) (red, basal body) and Lotus Tetragonolobus Lectin (LTL) (grey, proximal tubule). White arrows indicate primary cilia. Slides were imaged with a Zeiss LSM 980 confocal microscope (×63 objective, 2× digital zoom), scale bar represents 20 μm. All images are representative of $n = 4$ animals per experimental group. (B) Proximal tubule primary cilia length was measured using Imaris software. Four images were taken per an animal, 574 primary cilia were measured in the control and 712 primary cilia were measured in the GR-null mouse kidneys. Lines represent median and quartiles. All data presented as mean ± SEM, significant differences were analysed by unpaired T tests indicated by *$P ≤ 0.05$, **$P ≤ 0.01$, ***$P ≤ 0.001$, ****$P ≤ 0.0001$, ns=not significant, between control and GR-null ($P = 0.0001$), $n = 4$ animals per experimental group. (C) Percentage of proximal tubule primary cilia greater than 5 μm in control and GR-null proximal tubules. All data presented as mean ± SEM, significant differences were analysed by unpaired T tests indicated by *$P ≤ 0.05$, **$P ≤ 0.01$, ***$P ≤ 0.001$, ****$P ≤ 0.0001$, ns=not significant, between control and GR-null ($P = 0.027$), $n = 4$ animals per experimental group. (D) Percentage of ciliated proximal tubule cells. All data presented as mean ± SEM, significant differences were analysed by unpaired T tests indicated by *$P ≤ 0.05$, **$P ≤ 0.01$, ***$P ≤ 0.001$, ****$P ≤ 0.0001$, ns=not significant, between control and GR-null ($P = 0.11$), $n = 4$ animals per experimental group. (E) Scanning electron microscopy of primary cilia morphology in control and GR-null kidneys at E18.5. Kidneys were imaged with Nova NanoSEM 450 scanning electron microscope (Thermo Fisher Scientific) at a voltage of 2 kV and a spot size of 2. Black arrows indicate primary cilia. Scale bar represents 4 μm (control) and 5 μm (GR-null). All images are representative of $n = 5$ animals per experimental group. Source data are available online for this figure.

5 μm in GR-null kidney podocytes (7.25 ± 2.04%) compared to controls (12.2 ± 3.48%) (Fig. 5C) and no significant difference in the percentage of ciliated podocyte cells between GR-null kidney podocytes (25.64 ± 3.48%) and controls (24.15 ± 2.06%) (Fig. 5D).

## Dexamethasone increases the number and length of primary cilia in human kidney organoids

To further explore glucocorticoid regulation of primary ciliogenesis in a human cell-based model, induced pluripotent stem cell-derived kidney organoids were treated for 48 h with the synthetic glucocorticoid dexamethasone. Primary cilia were visualised by immunofluorescence as above (Fig. 5E). After dexamethasone treatment primary cilia were significantly longer (1.77 ± 0.02 μm) compared to controls (1.59 ± 0.02 μm) (Fig. 5F). Additionally, the percentage of primary cilia longer than 4 μm was significantly greater in podocytes for dexamethasone-treated organoids (2.80 ± 0.32%) compared to controls (0.9 ± 0.24%) (Fig. 5G). There were no significant differences in the percentage of ciliated cells in human kidney organoids treated with dexamethasone (76.62 ± 4.99%) versus controls (76.51 ± 2.75%) (Fig. 5H).

## Dexamethasone increased primary cilia length on IMCD3 cells and was blocked by the GR antagonist RU486

The effect of dexamethasone was next investigated with cultured mouse IMCD3 cells. Prior to steroid treatment IMCD3 cells were grown in charcoal-stripped media to reduce endogenous steroid hormones followed by serum starved media to induce ciliogenesis (Fig. EV5D). Primary cilia morphology was visualised with immunofluorescence as above after 96 h of vehicle or dexamethasone treatment (Fig. 6A). Primary cilia length in IMCD3 cells treated with dexamethasone (2.89 ± 0.04 μm) was significantly longer than vehicle (2.46 ± 0.03 μm) (Fig. 6B). Furthermore, the percentage of primary cilia longer than 4 μm was significantly higher in dexamethasone treated cells (54.56 ± 1.80%) versus controls (43.85 ± 1.30%) (Fig. 6C), with no significant difference in the percentage of ciliated collecting duct cells in cells following either treatment (Fig. 6D). To determine whether primary cilia elongation was in response to dexamethasone acting via the GR, IMCD3 cells were treated with $10^{-6}$ M RU486, a high affinity GR antagonist. Primary cilia morphology in IMCD3 cells after RU486 and then 96 h of dexamethasone treatment was visualised as above.

Primary cilia length on IMCD3 cells treated with dexamethasone (2.89 ± 0.04 μm) was significantly longer than vehicle (2.46 ± 0.03 μm) and dexamethasone + RU486 (2.00 ± 0.02 μm), indicating that RU486 blocked the effect of dexamethasone acting via the GR (Fig. 6B). In addition, the percentage of primary cilia longer than 4 μm was significantly higher in cells treated with dexamethasone (21.78 ± 1.68%), compared to all other treatments (vehicle: 11.17 ± 1.70%, vehicle + RU486: 5.50 ± 0.10% and dexamethasone + RU486: 6.12 ± 0.88%) (Fig. 6C). There were no significant differences in the percentage of ciliated collecting duct cells, except between dexamethasone (65.77 ± 2.87%) and dexamethasone + RU486 (77.43 ± 2.29%) (Fig. 6D).

## GR-null mice have reduced P-ERK and SUFU protein levels and dexamethasone treatment reduces Aurora kinase A protein levels in IMCD3 cells

Gene set enrichment analysis identified several signaling pathway genes that were differentially expressed in GR-null fetal kidney when compared to control (Fig. EV3A). It has been demonstrated that several WNT signaling proteins are upregulated in the GR-null lung (Bridges et al, 2020), suggesting that primary ciliogenesis which normally occurs during the $G_0$ quiescent phase of the cell cycle, could potentially be controlled by developmental and proliferation pathways. To explore the mechanism of glucocorticoid regulation of primary cilia, western blot analysis was used to quantify several signaling pathways and ciliary proteins (Fig. 7A–L). There was no significant difference in protein levels and phosphorylation of AKT (Fig. 7A), AMPKα (Fig. 7B), β-catenin (Fig. 7C), JNK (Fig. 7E) and S6 kinase (Fig. 7F) signalling proteins between control and GR-null mice. P-ERK and total ERK protein levels were significantly decreased in GR-null mice (0.53 ± 0.04) compared to controls (0.81 ± 0.09) (Fig. 7D). Suppressor of Fused (SUFU) protein levels were significantly reduced in GR-null mice (0.81 ± 0.06) compared to controls (0.95 ± 0.02) (Fig. 7G). There were no significant differences in the levels of ciliary structural proteins acetylated tubulin (Fig. 7I), CEP290 (Fig. 7J), IFT88 (Fig. 7K) and KIF3A (Fig. 7L) between control and GR-null mice. Aurora kinase A (AURKA) has been shown to promote cilia resorption (Pugacheva et al, 2007) and so we next examined whether AURKA may be regulated by the GR. IMCD3 cells were treated with dexamethasone and reduced levels of AURKA protein were observed (0.43 ± 0.03) compared to control cells (0.75 ± 0.09),

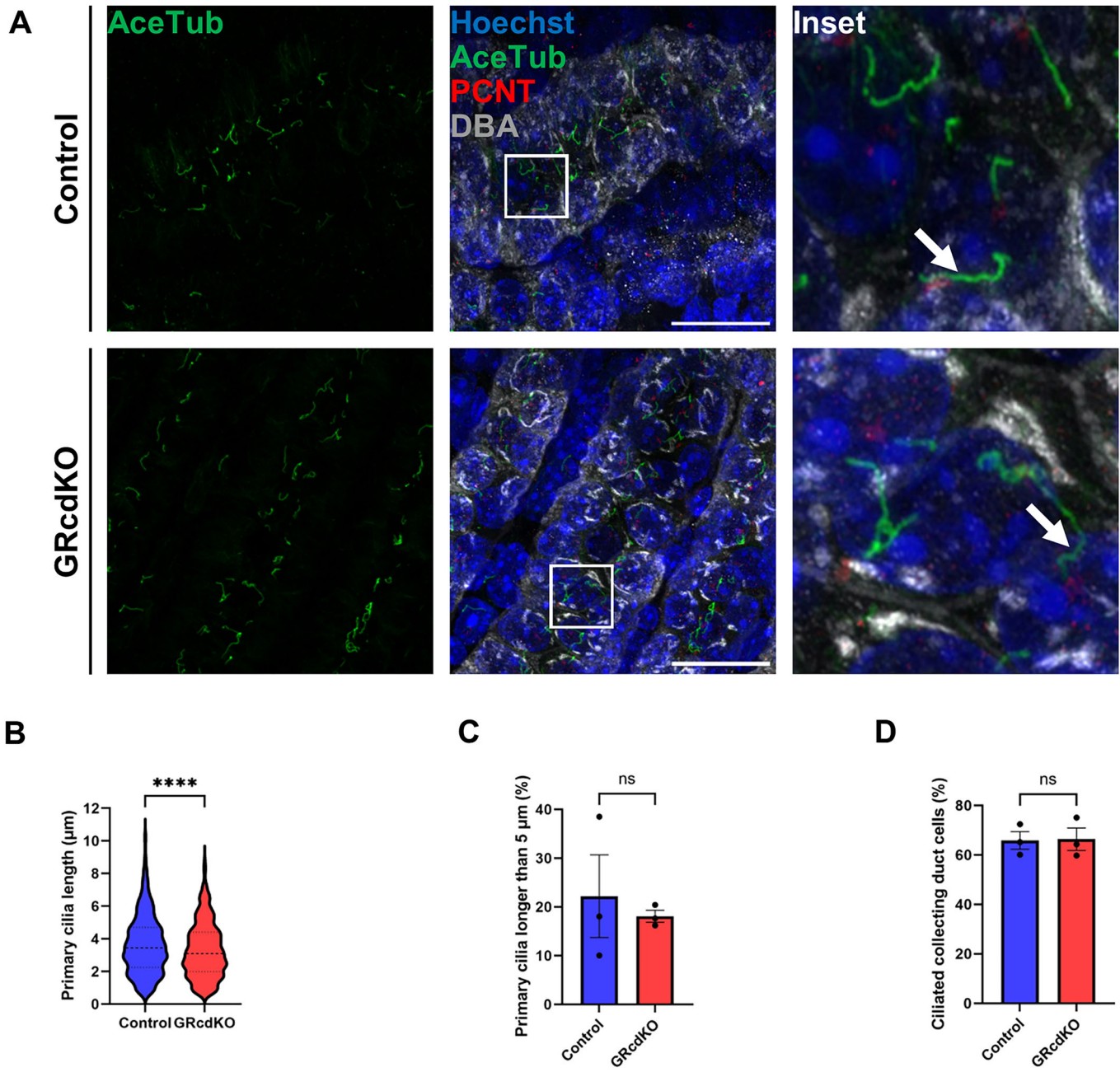

**Figure 4. Glucocorticoid regulation of primary cilia length on GRcdKO postnatal kidney collecting duct cells at P11.**

(**A**) Immunofluorescence of primary cilia morphology in control and GRcdKO collecting ducts at P11. Sections were stained with Hoechst (blue, nucleus), acetylated tubulin (AceTub) (green, microtubules), pericentrin (PCNT) (red, basal body) and Dolichos Biflorus Agglutinin (DBA) (grey, collecting duct). White arrows indicate primary cilia. Slides were imaged with a Zeiss LSM 980 confocal microscope (×63 objective, 2× digital zoom), scale bar represents 20 μm. All images are representative of $n = 4$ animals per experimental group. (**B**) Collecting duct primary cilia length was measured using Imaris software. Four images were taken per an animal, 1048 primary cilia were measured in the control and 957 primary cilia were measured in the GRcdKO mouse kidneys. Lines represent median and quartiles. All data presented as mean ± SEM, significant differences were analysed by unpaired T tests indicated by *$P ≤ 0.05$, **$P ≤ 0.01$, ***$P ≤ 0.001$, ****$P ≤ 0.0001$, ns=not significant, between control vs GRcdKO ($P = 0.0001$), $n = 3$ animals per experimental group. (**C**) Percentage of collecting duct primary cilia greater than 5 μm in the control and GRcdKO mouse kidney at P11. All data presented as mean ± SEM, significant differences were analysed by unpaired T tests indicated by *$P ≤ 0.05$, **$P ≤ 0.01$, ***$P ≤ 0.001$, ****$P ≤ 0.0001$, ns=not significant, between control and GRcdKO ($P = 0.66$), $n = 3$ animals per experimental group. (**D**) Percentage of ciliated collecting duct cells. All data presented as mean ± SEM, significant differences were analysed by unpaired T tests indicated by *$P ≤ 0.05$, **$P ≤ 0.01$, ***$P ≤ 0.001$, ****$P ≤ 0.0001$, ns=not significant, between control and GRcdKO ($P = 0.94$), $n = 3$ animals per experimental group. Source data are available online for this figure.

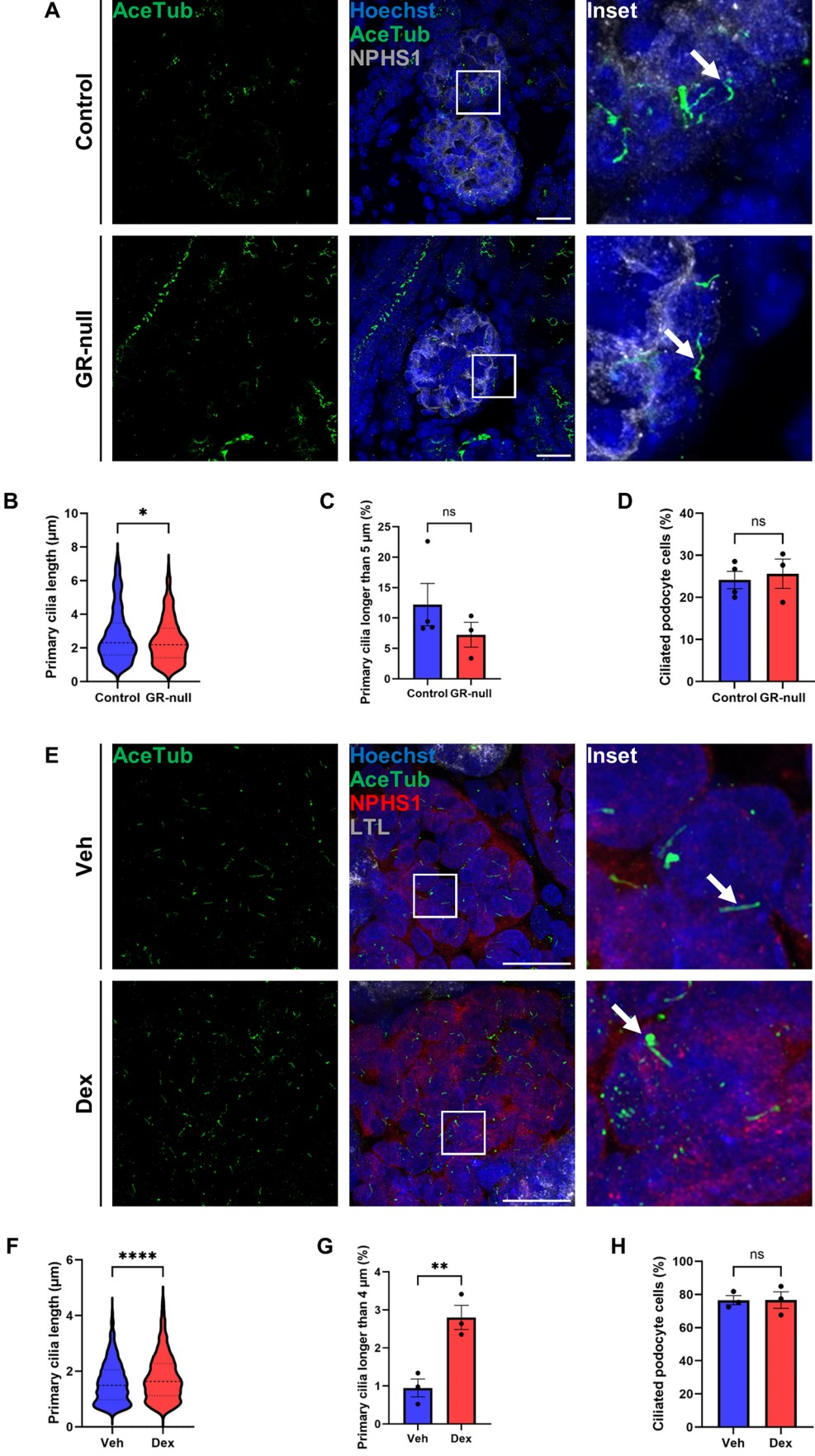

**Figure 5. Glucocorticoid regulation of primary cilia length on GR-null fetal kidney podocyte cells at E18.5 and induced pluripotent stem cell kidney organoid podocyte cells after dexamethasone treatment.**

(A) Immunofluorescence of primary cilia morphology in control and GR-null podocytes at E18.5. Sections were stained with Hoechst (blue, nucleus), acetylated tubulin (AceTub) (green, microtubules) and Nephrin (NPHS1) (grey, podocyte). White arrows indicate primary cilia. Slides were imaged with a Zeiss LSM 980 confocal microscope (×63 objective, 2× digital zoom), scale bar represents 20 µm. All images are representative of $n = 4$ animals per experimental group. (B) GR-null podocyte primary cilia length was measured using Imaris software. Four images were taken per animal, 235 primary cilia were measured in control and 212 primary cilia were measured in GR-null. Lines represent median and quartiles. All data presented as mean ± SEM, significant differences were analysed by unpaired T tests indicated by *$P \le 0.05$, **$P \le 0.01$, ***$P \le 0.001$, ****$P \le 0.0001$, ns=not significant, between control and GR-null ($P = 0.030$). Data from control ($n = 4$) and GR-null ($n = 3$) animals per experimental group. (C) Percentage of primary cilia greater than 5 µm in control and GR-null podocyte cells. All data presented as mean ± SEM, significant differences were analysed by unpaired T tests indicated by *$P \le 0.05$, **$P \le 0.01$, ***$P \le 0.001$, ****$P \le 0.0001$, ns=not significant, between control and GR-null ($P = 0.32$). Data from control ($n = 4$) and GR-null ($n = 3$) animals per experimental group. (D) Percentage of ciliated podocyte cells. All data presented as mean ± SEM, significant differences were analysed by unpaired T tests indicated by *$P \le 0.05$, **$P \le 0.01$, ***$P \le 0.001$, ****$P \le 0.0001$, ns=not significant, between control and GR-null ($P = 0.71$). Data from control ($n = 4$) and GR-null ($n = 3$) animals per experimental group. (E) Immunofluorescence of primary cilia morphology in induced pluripotent stem cell (iPSC) kidney organoids. iPSC kidney organoids were treated with vehicle (veh) or dexamethasone (dex). Sections were stained with Hoechst (blue, nucleus), acetylated tubulin (AceTub) (green, microtubules), nephrin (NPHS1) (red, podocyte) and Lotus Tetragonolobus Lectin (LTL) (grey, proximal tube). White arrows indicate primary cilia. Slides were imaged with a Zeiss LSM 980 confocal microscope (×63 objective, 2× digital zoom), scale bar represents 20 µm. All images are representative of $n = 4$ animals per experimental group. (F) Podocyte primary cilia length was measured using Imaris software. Four images were taken per animal, 1685 primary cilia were measured in vehicle (veh) treated organoids and 1638 primary cilia were measured in dexamethasone (dex) treated organoids. Lines represent median and quartiles. All data presented as mean ± SEM, significant differences were analysed by unpaired T tests indicated by *$P \le 0.05$, **$P \le 0.01$, ***$P \le 0.001$, ****$P \le 0.0001$, ns=not significant, between veh and dex treated cells ($P = 0.0001$), $n = 3$ biological replicates per experimental group. (G) Percentage of primary cilia greater than 4 µm in veh and dex treated organoids. All data presented as mean ± SEM, significant differences were analysed by unpaired T tests indicated by *$P \le 0.05$, **$P \le 0.01$, ***$P \le 0.001$, ****$P \le 0.0001$, ns=not significant, between veh and dex treated cells ($P = 0.0093$), $n = 3$ biological replicates per experimental group. (H) Percentage of ciliated podocyte cells. All data presented as mean ± SEM, significant differences were analysed by unpaired T tests indicated by *$P \le 0.05$, **$P \le 0.01$, ***$P \le 0.001$, ****$P \le 0.0001$, ns=not significant, between veh and dex treated cells ($P = 0.99$), $n = 3$ biological replicates per experimental group. Source data are available online for this figure.

thereby confirming AURKA expression is negatively regulated by GR activity (Fig. 7H).

## Discussion

This study has investigated GC/GR signalling in the mouse fetal kidney and demonstrates a key regulatory role for GC signalling in primary ciliogenesis. We first showed that the GR is expressed at E13.5 within nephron progenitor, collecting duct, stromal and immune cell populations, and expands into almost all cell types within the fetal mouse kidney where it has the capacity to respond to the rising levels of late gestational endogenous corticosterone from E15-E16 prior to birth (Fowden and Forhead, 2015). To further investigate the role of GC signalling in the developing kidney prior to birth we utilised mice deficient in the GR. Loss of GR in the fetal kidney had a substantial effect on the renal transcriptome, with 288 genes differently expressed more than 1-fold at E18.5. The majority of these genes were significantly decreased in expression including the kidney tubule markers *Kap* and *S100g*. *Kap* is one of the most abundant genes expressed in the kidney proximal tubule cells and is regulated by androgens however the role and function of *Kap* in proximal tubule cells is unknown (Teixido et al, 2006). One study by de Quixano et al showed that overexpression of *Kap* protected mice from metabolic syndrome induced by a high fat diet including associated hypertension (de Quixano et al, 2017). *S100g*, is a calcium-binding protein localised to the distal tubule at E18.5. Decreased *S100g* may indicate that GC signalling is involved in calcium transport within the kidney. However, *S100g* knockout mice are indistinguishable from control mice (Kutuzova et al, 2006; Lee et al, 2007). This may be a result of compensation from other calcium transporter proteins.

Of the ciliogenesis-related genes, *Cep290* was strongly decreased in the mouse kidney of GR-null fetal mice at E18.5. Although we detect similar levels of *Cep290* protein in total fetal kidney extracts,

further studies are required with isolated primary kidney epithelial cells to specifically assess Cep290 protein levels in ciliated renal tubular cells. Other structural cilia proteins, such as *Ccp110* and *Cep97*, also need to be assessed. The CEP290 protein is localised to the transition zone of primary cilia where it is essential for assembly of microtubules and for primary cilia formation. *Cep290* genetic mutations are found in a number of human ciliopathies including Joubert syndrome, characterised by brain abnormalities and in some instance's polycystic kidneys, as well as Meckel syndrome which is characterised by cystic dysplastic kidneys, brain malformations, hepatic fibrosis and proliferation of the bile ducts. This commonly leads to end-stage renal disease by the age of 30 and is characterised by cystic kidneys, and finally Senior-Loken syndrome characterised by kidney cysts, inflammation and scarring leading to end stage kidney disease (Baala et al, 2007; Helou et al, 2007; Hildebrandt et al, 2011; Valente et al, 2006). Furthermore, mice deficient in *Cep290* develop symptoms consistent with ciliopathies including reduced numbers of primary cilia on renal epithelial cells and the development of kidney cysts (Rachel et al, 2015). Development of kidney cysts in *Cep290* knockout mice is progressive and cysts become more prominent at 2 weeks of age (Rachel et al, 2015). Unfortunately, due to the early lethality of GR-null mice we are unable to determine if global GR-null mice develop kidney cysts. Five other cilia-associated genes including *Ccp110, Cep97, Kif3a* and *Rpgr* were also significantly down-regulated in the GR-null mouse kidney at E18.5. Finally, we also assessed protein levels of two other cilia-related structural proteins, IFT88 and acetylated tubulin with no significant change in their protein level detected by western blot analysis in the E18.5 fetal kidney of GR-null mice. This likely either reflects no direct regulation of their expression by GR-mediated cell signalling or no direct role in the mechanism causing the cilia defect in these mice. Future analysis in isolated primary kidney tubule epithelial cells could be used to rule out a mechanistic role here for IFT88 and acetylated tubulin, and also other cilia-related structural proteins.

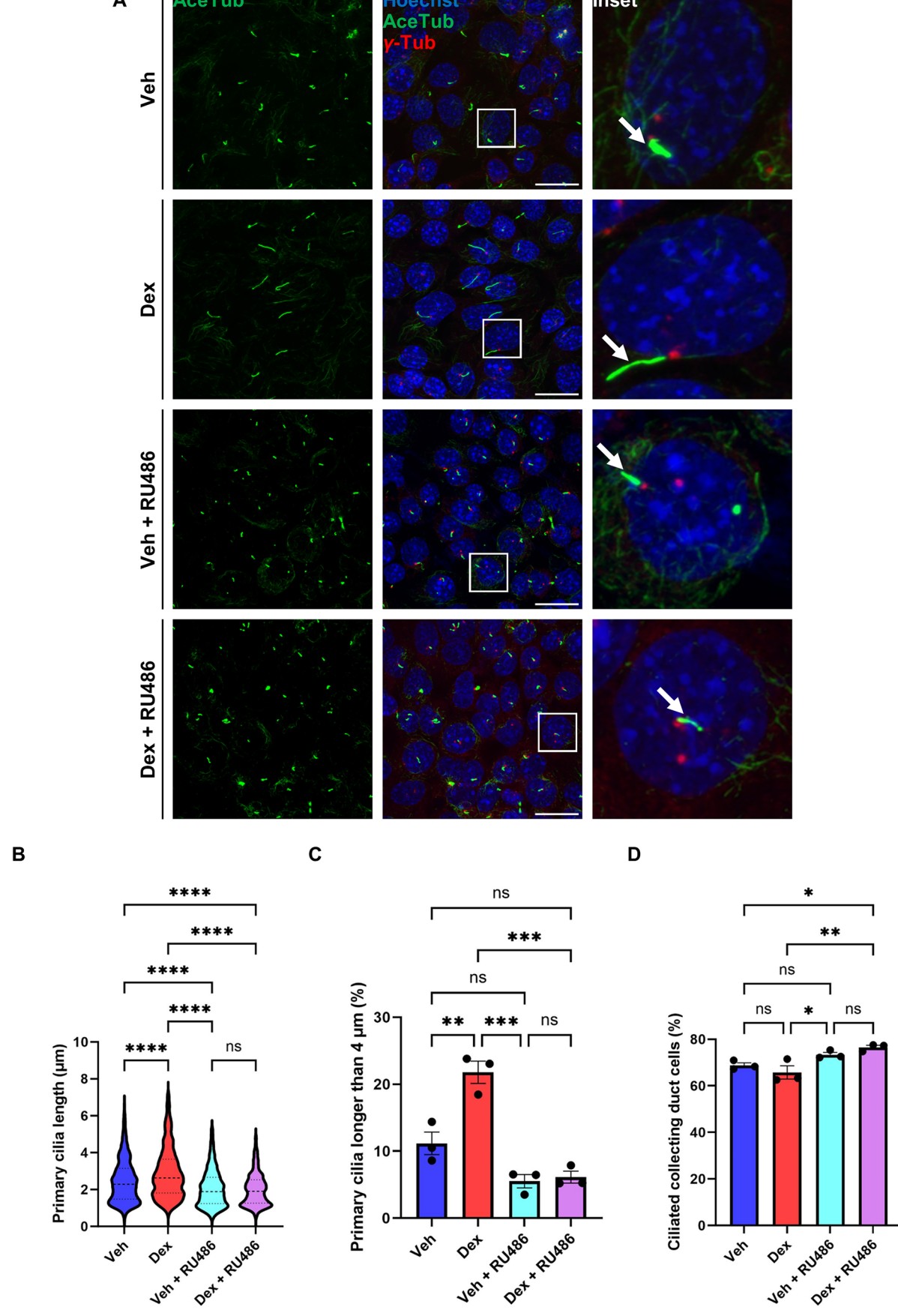

**Figure 6. Glucocorticoid regulation of primary cilia in IMCD3 cells after dexamethasone treatment.**

(A) Immunofluorescence images of primary cilia morphology in IMCD3 cells. IMCD3 cells were treated with vehicle (veh), dexamethasone (dex), vehicle + RU486 (veh + RU486) or dexamethasone + RU486 (dex + RU486). Sections were stained with Hoechst (blue, nucleus), acetylated tubulin (AceTub) (green, microtubules) and gamma tubulin (γ-Tub) (red, basal body). White arrows indicate primary cilia. Slides were imaged with a Lecia SP8 confocal microscope (×63 objective, 2× digital zoom), scale bar represents 20 μm. All images represent $n = 3$ biological replicates per experimental group. (B) Primary cilia length was measured using Imaris software. Four images were taken per biological replicate, the number of individual cilia measured per a treatment group was 1698 vehicle, 1514 dexamethasone, 1966 vehicle + RU486 and 2143 dexamethasone + RU486. Lines represent median and quartiles. All data presented as mean ± SEM, significant differences were analysed by one-way ANOVA with multiple comparisons indicated by *$P \le 0.05$, **$P \le 0.01$, ***$P \le 0.001$, ****$P \le 0.0001$, ns=not significant. Veh vs dex ($P = 0.0001$), veh vs veh + RU486 ($P = 0.0001$), veh vs dex + RU486 ($P = 0.0001$), dex vs veh + RU486 ($P = 0.0001$), dex vs dex + RU486 ($P = 0.0001$), veh + RU486 vs dex + RU486 ($P = 0.28$). Data from $n = 3$ biological replicates per experimental group. (C) Percentage of primary cilia greater than 4 μm. All data presented as mean ± SEM, significant differences were analysed by one-way ANOVA with multiple comparisons indicated by *$P \le 0.05$, **$P \le 0.01$, ***$P \le 0.001$, ****$P \le 0.0001$, ns=not significant. Veh vs dex ($P = 0.0026$), veh vs veh + RU486 ($P = 0.073$), veh vs dex + RU486 ($P = 0.11$), dex vs veh RU486 ($P = 0.0001$), dex vs dex + RU486 ($P = 0.0002$), veh + RU486 vs dex + RU486 ($P = 0.99$). Data from $n = 3$ biological replicates per experimental group. (D) Percentage of ciliated collecting duct cells. All data presented as mean ± SEM, significant differences were analysed by one-way ANOVA with multiple comparisons indicated by *$P \le 0.05$, **$P \le 0.01$, ***$P \le 0.001$, ****$P \le 0.0001$, ns=not significant. Veh vs dex ($P = 0.59$), veh vs veh + RU486 ($P = 0.29$), veh vs dex + RU486 ($P = 0.046$), dex vs veh + RU486 ($P = 0.049$), dex vs dex + RU486 ($P = 0.0081$), veh + RU486 vs dex + RU486 ($P = 0.57$). Data from $n = 3$ biological replicates per experimental group. Source data are available online for this figure.

Analysis of cilia structure at E18.5 in proximal tubule cells by both fluorescent confocal microscopy and scanning electron microscopy showed shortened and abnormally shaped primary cilia in renal tubules from GR-null mice. Strikingly, this phenotype was replicated in renal collecting ducts postnatally in collecting duct-specific-GR-null mice. In contrast, treatment of the renal IMCD3 collecting duct cell line with dexamethasone, a synthetic GC agonist, drove increased primary cilia growth that was inhibited by pre-treatment with the GR antagonist RU486. Dexamethasone stimulation of cilia length was also replicated in iPSC-derived kidney organoids supporting a conserved role for GC/GR signalling in human kidney cell types. Further studies using human kidney organoids co-labelling specific renal tubular cells will clarify if these regulatory effects on cilia length occur in all human tubule cells or are restricted to specific tubular segments such as the proximal or distal tubules. Analysis of key ciliogenesis signalling pathways showed reduced ERK and SUFU activity in GR-null fetal kidney extracts and repression by dexamethasone of an important promoter of cilia disassembly, AURKA, in the IMCD3 cell line (Pugacheva et al, 2007). However, cystogenesis is not as simple as stabilising or destabilising the primary cilia. While *Kif3a* (Lin et al, 2003) and *Cep164* (Airik et al, 2019) deletion can cause cystogenesis due to a lack of primary cilia in neonatal contexts, in adult models of autosomal dominant PKD, Ma et al have demonstrated a cilia-dependent cyst activation pathway (CDCA) where a structurally intact cilia transmits an aberrant growth signal which cilia removal via *Kif3a* co-deletion prevents (Ma et al, 2017). Hence, there are likely multiple cystogenic pathways associated with cilia, such as the major and minor pathways proposed by Hwang et al (Hwang et al, 2019).

Apart from the single case report that described enhanced survival of a preterm infant with advanced PKD following treatment with dexamethasone (Katz et al, 1979), no clinical studies have been reported directly using glucocorticoid steroids to treat PKD (Forcioli-Conti et al, 2015; Katz et al, 1979). In summary, these results clearly define glucocorticoid signalling as an important regulator and driver of primary cilia formation in renal proximal epithelial cells. The exact mechanism of cilia shortening is not clear and may involve a defect in formation or maintenance of correct cilium length or perhaps increased shedding of cilia. Scanning electron microscopy images of fetal kidney cells from GR-null mice (Fig. 3) show the presence of abnormally shaped bulging cilia that

could contribute to cilia shortening, but this requires further more detailed analysis. Our results complement the report showing that another steroid hormone aldosterone, acting via the mineralocorticoid receptor, also controls primary cilia growth and length in collecting duct cells (Komarynets et al, 2020). Glucocorticoid signalling via genomic actions positively regulate an important subset of ciliogenesis genes, such as *Cep290*, *Cep110* and *Kif3a* to promote normal cilia growth and structure. Non-genomic effects where the GR may antagonise other intracellular signalling pathways may also contribute to cilia formation under high endogenous glucocorticoid actions or delivery of exogenous synthetic agonists. Non-genomic cytosolic interactions of the GR with a number of signalling kinases has been described and include RAS, JNK and PI3K where the interaction by GR contributes to anti-inflammatory or tumorigenic responses (Caelles et al, 1997; Caratti et al, 2022; Limbourg et al, 2002). A key regulator of ciliogenesis is AURKA, a kinase that promotes cell proliferation and negatively regulates ciliogenesis in the kidney. Deletion of AURKA in mouse models of autosomal dominant PKD and Joubert Syndrome reduces cyst formation and the disease severity of PKD, in part via modulating AKT signalling in the kidney (Tham et al, 2024). We observed reduced AURKA protein levels in dexamethasone treated renal collecting duct cells, indicating the glucocorticoid signalling via the GR regulates AURKA during kidney development, which could be either direct or indirect. We did not see mRNA changes for *Aurka* in GR-null mice (Fig. 2F), so we predict this may be post-translational mechanism that involves a direct interaction between the two proteins. In fact in a recent study, Qiao et al demonstrated that the bone osteoclast lineage, AURKA interacts with the GR to prevent glucocorticoid-induced bone loss (Qiao et al, 2023). Compellingly in another paper by Sun et al studying mechanisms in ovarian endometriosis, AURKA has also been shown to interact with a related steroid hormone receptor, ERβ, where AURKA promotes a state of cell proliferation and invasion (Sun et al, 2024). Further studies are required to confirm direct interaction of the GR with AURKA in ciliated cells and the effect this has on signalling cascades. Finally, this study provides a rationale for the potential use of synthetic glucocorticoid agonists, such as dexamethasone or prednisolone, as a treatment option of human ciliopathies.

Primary cilia are cellular and environment sensing organelles that protrude from cell membranes and have important roles

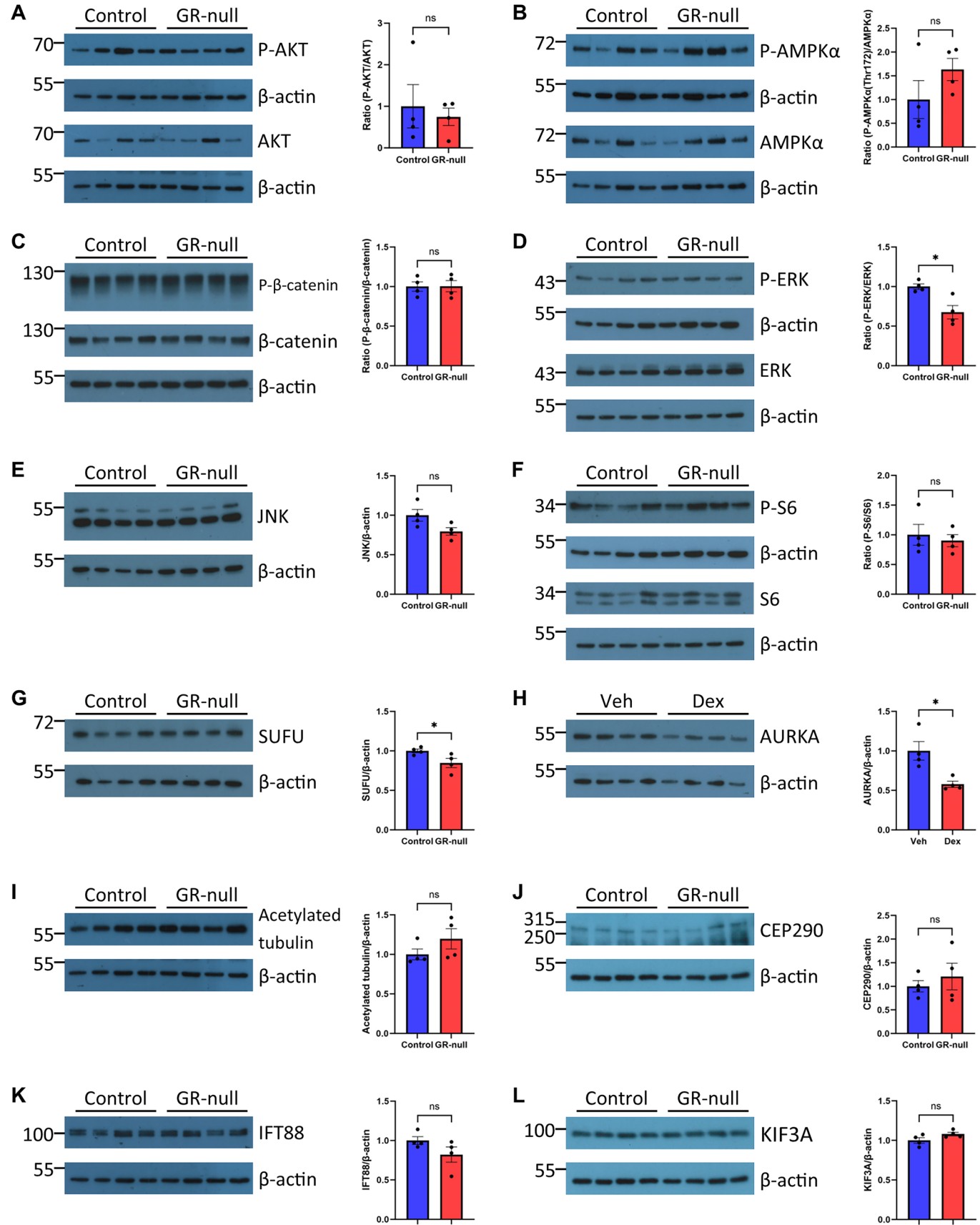

◄ **Figure 7. Analysis of cell signalling pathways and cilia structural proteins by western blot analysis as targets for glucocorticoid regulation of primary ciliogenesis.**

Western blot analysis of signaling pathway proteins (**A**) P-AKT and AKT ($P = 0.067$), (**B**) P-AMPKα and AMPKα ($P = 0.22$), (**C**) P-β-catenin and β-catenin ($P = 0.96$), (**D**) P-ERK and ERK ($P = 0.022$), (**E**) JNK ($P = 0.06$), (**F**) P-S6 and S6 ($P = 0.64$), and (**G**) SUFU ($P = 0.049$) in control and GR-null fetal kidney at E18.5. Western blot analysis of (**H**) Aurora kinase A (AURKA) ($P = 0.017$) in IMCD3 cells treated with vehicle (veh) or dexamethasone (dex). Western blot analysis of cilia structural proteins (**I**) acetylated tubulin ($P = 0.090$), (**J**) CEP290 ($P = 0.52$), (**K**) IFT88 ($P = 0.15$), (**L**) KIF3A ($P = 0.34$) in control and GR-null fetal kidney at E18.5. All data presented as mean ± SEM, significant differences were analysed by unpaired T tests indicated by *$P \le 0.05$, **$P \le 0.01$, ***$P \le 0.001$, ****$P \le 0.0001$, ns=not significant, between control and GR-null, $n = 4$ animals per experimental group or veh and dex, $n = 3$ biological replicates per experimental group. Western blots from P-AKT and AKT (**A**) were re-probed with IFT88 (**K**) and acetylated tubulin (**I**), respectively, and share the same β-actin blot. Western blots from P-ERK and ERK (**D**) was re-probed with P-S6 and S6 (**F**), respectively, and share the same β-actin blot. KIF3A (**L**) was re-probed with CEP290 (**J**) and share the same β-actin blot. The same β-actin panels were included in each figure for ease of comparison. Source data are available online for this figure.

during embryogenesis and tissue homeostasis. Defects in cilia cause many human ciliopathies including nephronophthisis and polycystic kidney disease. We show that glucocorticoid steroids action in vivo and in vitro via the cellular glucocorticoid receptor regulate and enhance normal ciliogenesis and cilia growth. Our data demonstrate that glucocorticoid signalling is required for normal primary ciliogenesis.

## Methods

### Reagents and tools table

| Experimental models | Reference or source | Identifier or catalogue number |
|---|---|---|
| GRcdKO, C57BL6/J (*M. musculus*) | This study | N/A |
| GR-null, C57BL6/J (*M. musculus*) | Cole et al, 1995 | N/A |
| mIMCD3 cells (*M. musculus*) | American Type Culture Collection | CRL-2123 |
| Induce pluripotent stem cell kidney organoid, line 522.3 (*H. sapiens*) | GM04522 | NIGMS Human Genetic Cell Repository |
| **Primary antibodies** | **Reference or source** | **Identifier or catalogue number** |
| Akt rabbit polyclonal | Cell Signaling Technology | 9272 |
| AMPKα rabbit monoclonal | Cell Signaling Technology | 5832 |
| Ki67 rabbit polyclonal | Abcam | Ab15580 |
| Pericentrin rabbit polyclonal | Abcam | Ab4448 |
| CEP290 rabbit polyclonal | Novus Biologicals | NB100-86991 |
| Dolichos Biflorus Agglutinin (DBA), biotinylated | Vector Laboratories | B-1035 |
| Glucocorticoid receptor rabbit monoclonal | Cell Signaling Technology | 12041 |
| Hoechst 33342, trihydrochloride trihydrate | Invitrogen | H1399 |
| Nephrin sheep polyclonal | R&D Systems | AF4269 |

| Experimental models | Reference or source | Identifier or catalogue number |
|---|---|---|
| IFT88 rabbit polyclonal | Proteintech | 13967-1-AP |
| JNK1 (2C6) mouse monoclonal | Cell Signaling Technology | 3708 |
| KIF3A rabbit polyclonal | GeneTex | GTX134434 |
| Lotus Tetragonolobus Lectin (LTL), biotinylated | Vector Laboratories | B-1325 |
| MEIS 1/2/3 mouse monoclonal | Active Motif | 39795 |
| Acetylated tubulin mouse monoclonal | Sigma-Aldrich | T6793 |
| β-actin mouse monoclonal | Sigma-Aldrich | A5441 |
| p44/42 MAPK (ERK1/2) rabbit monoclonal | Cell Signaling Technology | 4695 |
| P-Akt (S473) rabbit monoclonal | Cell Signaling Technology | 4060 |
| Phospho-p44/42 MAPK (Erk1/2) (Thr202/Tyr204) rabbit monoclonal | Cell Signaling Technology | 9101 |
| Phospho-AMPKα (Thr172) rabbit monoclonal | Cell Signaling Technology | 50081 |
| Phospho-S6 Ribosomal Protein (Ser235/236) rabbit polyclonal | Cell Signaling Technology | 2211 |
| Phospho-β-Catenin (Ser675) rabbit monoclonal | Cell Signaling Technology | 4176 |
| IAK1 (Aurka) mouse monoclonal | BD Biosciences | 610939 |
| Gamma Tubulin rabbit monoclonal | Abcam | Ab179503 |
| S6 Ribosomal Protein mouse monoclonal | Cell Signaling Technology | 2317 |
| SUFU rabbit monoclonal | Cell Signaling Technology | 2522 |
| β-Catenin rabbit monoclonal | Cell Signaling Technology | 8480 |

| Experimental models | Reference or source | Identifier or catalogue number |
|---|---|---|
| **Secondary antibodies** | **Reference or source** | **Identifier or catalogue number** |
| Anti-mouse, HRP-linked Antibody | Cell Signaling Technology | 7076 |
| Anti-rabbit, HRP-linked Antibody | Cell Signaling Technology | 7074 |
| Donkey anti-Goat, Alexa Fluor 647 | Invitrogen | A32849 |
| Donkey anti-Mouse, Alexa Fluor 555 | Invitrogen | A31570 |
| Donkey anti-Mouse, Alexa Fluor Plus 647 | Invitrogen | A32787 |
| Donkey anti-Rabbit, Alexa Fluor 488 | Invitrogen | A21206 |
| Donkey anti-Rabbit, Alexa Fluor 568 | Invitrogen | A10042 |
| Donkey anti-Rabbit, Alexa Fluor 647 | Invitrogen | A31573 |
| Donkey anti-Sheep, Alexa Fluor 647 | Invitrogen | A21448 |
| Streptavidin, Alexa Fluor 488 Conjugate | Thermo Fisher Scientific | S32354 |
| **qRT-PCR primers** | | |
| Gene | Forward primer 5′ | Reverse primer 5′ |
| Atat1 | TGTTACAGAAAGAGCGAGTGGA | TGGTCTCCAGGTTGTAGTGC |
| Aurka | TCTCGGGTTGAATTCACTTTCC | CCTTTGGCTTGCGTTGTGTT |
| Ccp110 | CCGTGGCAAAAGGGTTTC | CTCGCAACTGAGCTAACAC |
| Cep97 | AGTCTGAAGGCTGAGTGG | ACTCTTCATCTTGGCTCTGG |
| Cep290 | AATGAAGATGAAAGCCCAGGA | TGCTTCTCAAGTTGGCGAAT |
| Crisp1 | TGGTCTTCTGCAATCCAAGG | TGAGTTCCAAACAACCTGAGT |
| Gbp7 | CAGCTGCACTATGTCACAG | CACGTCCCTCTAACTTCAGC |

| Experimental models | Reference or source | Identifier or catalogue number |
|---|---|---|
| Hba-a1 | AAGCCCTGGAAAGGATGTTT | CTCAGGAGCTTGAAGTTGAC |
| Ifi203 | CAGTGGTGGTTTATGGACGA | CTCAGGAGGCACACATCTTT |
| Ifit1 | GCTGTCCGGTTAAATCCAGA | AAGTAGCCAGAGGAAGGTGA |
| Igf2 | TCTCATCTCTTTGGCCTTCG | CAAACTAAGCGTGTCAACA |
| Kap | TTAAGCACTGACTCTGAGCA | ACTGTGATGTCTGTGTTCTCA |
| Kif3a | CGTGGATGAAATGAGGGGAA | CCCATTGTAGCCTTCCAGAA |
| Mndal | TCGTCAAGATCAAGGTCACC | ACGTGATAGTCTGGCATCTC |
| Nedd9 | GCGAGGAATCTTATGGCAAGG | TCAAGCCCTCCTGTGTTCTG |
| Oas2 | AGTCTACTCCGCCTGATCAA | TGTCAAAGTCATCTGTGCCA |
| Oasl2 | GATTAAGGTGGTGAAGGGAGG | CCTGCTCTTCGAAACTGGAA |
| Rab8a | GCGAAGACCTACGATTACCT | CCGTGTCCCATATCTGCA |
| Rpgr | CATCCGCTGCTCTTACTGA | GCTCCCCATCCATTGTTACA |
| Rps29 | GGACATAGGCTTCATTAAGTTGG | TCAGTCGAATCCATTCAAGGT |
| S100g | GAGCTGGATAAGAATGGCGA | TTCAGGATTGGAGAGCGTG |
| **Chemical, enzymes and other reagents** | **Reference or source** | **Identifier or catalogue number** |
| Accutase | Stemcell Technologies | 07920 |

| Experimental models | Reference or source | Identifier or catalogue number |
| --- | --- | --- |
| Charcoal, Dextran Coated | Sigma-Aldrich | C6241 |
| CHIR99021 | Stemcell Technologies | 100-1042 |
| Clarity Western ECL Substrate | Bio-Rad | 1705061 |
| cOmplete™, EDTA-free Protease Inhibitor Cocktail | Roche | 04693132001 |
| Dexamethasone | Sigma-Aldrich | D4902 |
| Donkey serum | Sigma-Aldrich | D9663 |
| Essential 8™ Medium | Thermo Fisher Scientific | A1517001 |
| E6 medium | Thermo Fisher Scientific | A1516401 |
| Fetal Bovine Serum, qualified, New Zealand | Gibco | A3160902 |
| FGF9 | In Vitro Technologies | RDS233FB025 |
| Fujifilm Medical X-ray Film Blue Sensitive Super RX-N | Fujifilm | 47410 19289 |
| GlutaMAX™ Supplement | Thermo Fisher Scientific | 35050061 |
| Heparin | Sigma-Aldrich | H3149 |
| Immobilon®-P PVDF Membrane | Merck | IPVH00010 |
| Matrigel | Corning | FAL35427 |
| Mifepristone | Sigma-Aldrich | M8046 |
| N,N,N',N'-Tetramethylethylenediamine | Sigma-Aldrich | T7024 |
| Penicillin-Streptomycin (5000 U/mL) | Thermo Fisher Scientific | 15070063 |
| PhosSTOP™ | Roche | 4906845001 |
| ProLong™ Gold Antifade Mountant | Invitrogen | P36930 |
| SP Bel-Art Flowmi 70 Micron Cell Strainers | Bel-Art Products | H13680-0070 |
| TGX Stain-Free™ FastCast™ Acrylamide Kit, 10% | Bio-Rad | 1610183 |
| Triton™ X-100 | Sigma-Aldrich | X100 |
| TRIzol™ Reagent | Invitrogen | 15596026 |
| TrypLE Select Enzyme | Thermo Fisher Scientific | 12605028 |
| Y-27632 (Dihydrochloride) | Stemcell Technologies | 72302 |
| **Software and packages** | **Reference or source** | |
| BIORAD 384w rtPCR software CFX manager | Bio-Rad | |
| CIBERSORTx | Steen et al, 2020 | |
| Cyclone | Scialdone et al, 2015 | |
| DAVID | Sherman et al, 2022 | |
| edgeR R package | Robinson et al, 2010 | |

| Experimental models | Reference or source | Identifier or catalogue number |
| --- | --- | --- |
| FIJI | https://imagej.net/software/fiji/ | |
| GraphPad Prism | https://www.graphpad.com/features | |
| Imaris | https://imaris.oxinst.com/ | |
| Scrublet | Wolock et al, 2019 | |
| Seurat (v3.1.4) | Butler et al, 2018; Stuart et al, 2019 | |
| STAR (v2.5.1b) | Dobin et al, 2013 | |
| **Microscopes** | | |
| Leica SP8 confocal microscopy | | |
| Nova NanoSEM 450 scanning electron microscope | | |
| ZEISS LSM 980 with Airyscan 2 | | |
| **Kits** | **Reference or source** | **Identifier or catalogue number** |
| 10x Chromium v3 kits | Millennium Science | PN-1000699 |
| QuantiNova SYBR Green PCR Kit | Qiagen | 208052 |
| QuantiTect Reverse Transcription Kit | Qiagen | 205311 |

## Mice

Use of mice was approved by the MARP-2 Animal Ethics Committee at Monash University. Global GR-null (Bird et al, 2014; Cole et al, 1995) and collecting duct-specific GR-null (GRcdKO) mice were all of an isogenic C57BL/6J genetic background. GRcdKO mice were generated with HoxB7 promoter-Cre mice crossed with GR-floxed allele mouse (Yu et al, 2002). Fetal kidneys at E18.5 were dissected from embryos and either snap frozen in liquid $N_2$ or fixed in 4% paraformaldehyde. Tail snips were collected for genotyping by qPCR (Short et al, 2020).

## Fetal kidney single cell (SC) data analysis

The E18.5 kidney-SC data was generated as previously described and is available at GEO (GSE108291) (Combes et al, 2019; Data ref: Combes et al, 2019). SC RNA sequencing at E13.5 and E15.5 utilised wild type embryonic kidneys. The kidneys were dissociated in 500 μL Accutase (Stemcell Technologies) at 37 °C for 6–8 min, gently agitated every 2 min then washed with cold PBS 0.05% bovine serum, pelleted by centrifugation ($400 \times g$, 5 min), and stored on ice. Samples were filtered with Flowmi Cell Strainers (70 μm, Bel-Art Products) and stained with DAPI before removal of dead cells by FACS (100 μm nozzle). Cell concentration was determined using a hemocytometer and adjusted prior to the generation of single cell libraries using 10x Chromium v3 kits. Sequencing data was processed using Cell Ranger (10x Genomics, v1.3.1,) and aligned to mm10 with STAR (v2.5.1b) (Dobin et al, 2013). Subsequent analysis was performed in the R statistical programming language using Seurat (v3.1.4) (Butler et al, 2018; Stuart et al, 2019). Quality control for the E13.5 and E15.5 datasets involved removing cells with <1000 genes, >20% mitochondrial gene content (E13.5); <1500 gene, >8% mitochondrial gene content

(E15.5). Doublets were identified and filtered out using Scrublet (Wolock et al, 2019) or with *HTODemux* function in Seurat. Cell cycle phase was predicted using either Cyclone (Scialdone et al, 2015) or Seurat's *CellCycleScoring* function. Cell cycle effects were regressed out and gene expression data normalised using *SCTransform* with default parameters. Following all quality control steps, the E13.5 dataset consisted of 19,252 genes and 4176 cells and the E15.5 dataset of 18,549 genes and 3294 cells. Cluster identity was determined by referencing top cluster marker genes (*FindAllMarkers*) to Combes et al previous analysis (Combes et al, 2019). The new E13.5 and E15.5 datasets reported in this study are available upon request.

## Total RNA isolation and NGS transcriptome sequencing

Total RNA was isolated from embryonic kidneys and IMCD3 cells using TRIzol™ reagent (Invitrogen, USA) according to the manufacturer's instructions. Total RNA was analysed using a Bioanalyzer 2100 (Agilent Technologies, USA) and Next generation RNA sequencing (NGS RNA-seq) was performed by Genewiz Biotechnology, Suzhou, China. RNA sequencing (20 million reads) was performed on the Illumina Hiseq platform, in a 2 ×150 bp paired-end format.

## RNA-sequencing analysis

The gene expression count matrix underwent preprocessing and differential expression analysis with the edgeR R package (Robinson et al, 2010). Genes with low expression levels were filtered out, and the resulting log-transformed counts per million (CPM) values were utilized to create heatmaps. Quasi-Likelihood was used for statistical test. Sample N3 (PCA plot) was added as the extra covariate to the design matrix additionally to control versus knockout groups. Differentially expressed genes (DEGs) with a false discovery rate (FDR) less than 0.05 were identified and employed for generating volcano plots. Additionally, enrichment analysis was conducted using the DAVID online tool ($P$ value < 0.05) (Sherman et al, 2022). RNA-seq datasets reported in this study are available at GEO (GSE290962).

## Deconvolution

A reference count gene expression matrix was created by selecting the top 5000 most variable genes and randomly choosing 50 cells for each cell type from E18 developing mouse kidney single-cell RNA-seq data. This reference matrix was then provided to an online tool called CIBERSORTx (Steen et al, 2020). Additionally, the complete bulk RNA-seq count matrix was also provided to the software in order to generate the signature matrix and produce deconvolution results. These results display the percentage of each cell type in all bulk samples in a tabular format.

## cDNA synthesis and quantitative real-time PCR

cDNA was synthesised with a QuantiTect RT kit (Qiagen) according to the manufacturer's instructions from the same fetal kidney RNA samples used for RNA sequencing. mRNA levels of *Atat1, Aurka, Ccp110, Cep97, Cep290, Crisp1, Gbp7, Hba-a1, Ifi203, Ifit1, Igf2, Kap, Kif3a, Mndal, Nedd9, Oas2, Oasl2, Rab8a, Rpgr* and

*S100g* were determined by qRT-PCR using QuantiNova® SYBR® green master mix (Qiagen) on a CFX384 Touch Real-Time PCR Detection System (Bio-Rad). Relative mRNA levels were normalised to the housekeeping gene Ribosomal protein 29 (*Rps29*) using the ΔΔCt method (Pfaffl, 2001). PCR products for each primer set (Reagents and tools table) were verified by a PCR melt-curve analysis and DNA sequencing. Differentially expressed ciliary genes from the RNA-seq data of GR-null mice were matched from the CiliaCarta (van Dam et al, 2019) compendium and selected for qRT-PCR analysis.

## Human induce pluripotent stem cell kidney organoids

Differentiation of human induced pluripotent stem cell (iPSC) derived kidney organoids were developed as previously described (Takasato et al, 2015) with minor changes. 80,000 iPSCs were seeded in Matrigel on a 6-well plate supplemented with Essential 8 medium (Thermo Fisher Scientific), 10 μM ROCK inhibitor Y-27632 (In Vitro Technologies) and 1% penicillin-streptomycin (pen-strep) (Thermo Fisher Scientific) on day 0. Media was changed to Essential 6 (E6) medium, supplemented with 4 μM CHIR99021 (In Vitro Technologies) and 1% pen-strep the following day. In total, 200 ng/mL of FGF9 (In Vitro Technologies) and 1 μg/mL of Heparin (Sigma-Aldrich) were added to the media on day 4. Fresh media was changed every 2 days. iPSCs were dissociated with TrypLE Select Enzyme (1×) (Thermo Fisher Scientific) and 150,000 cells were used to generate each 3D kidney organoid on day 7. Kidney organoids were treated with 4 μM CHIR99021 in E6 medium for 1 h and changed to E6 medium, supplemented with 200 ng/mL of FGF9, 1 μg/mL of Heparin and 1% pen-strep. Fresh media was changed every 2 days until day 13, where media was changed to E6 medium supplemented with 1% pen-strep. Kidney organoids were collected at day 20 for 48 h of vehicle (ethanol) $10^{-6}$ M or dexamethasone $10^{-6}$ M in serum free media to induce ciliogenesis.

## IMCD3 cell culture and drug treatments

Mouse inner medullary collecting duct (IMCD3) cells were maintained in DMEM: Nutrient Mixture F-12 (DMEM/F12), supplemented with 10% FBS (Gibco), 1% L-glutamine (Thermo Fisher Scientific) and 1% pen-strep (Thermo Fisher Scientific) in 5% $CO_2$ at 37 °C. Cells were incubated in media containing charcoal-stripped FBS for 16 h, and then treated with either vehicle (ethanol), dexamethasone $10^{-6}$ M, vehicle + RU486 $10^{-6}$ M, or dexamethasone $10^{-6}$ M + RU486 $10^{-6}$ M for 48 h. Cells were incubated in serum free media to induce ciliogenesis for an additional 48 h with either vehicle, dexamethasone, vehicle + RU486 or dexamethasone + RU486.

## Periodic acid-Schiff staining, immunofluorescence and primary cilia analysis

Fetal kidneys fixed in 4% paraformaldehyde were embedded in paraffin and cut at either 4 μm for periodic acid-Schiff (PAS) staining or 10 μm sections for immunofluorescence staining. PAS staining was performed on a Leica ST5010 Autostainer and CV5030 coverslipper and scanned with an Aperio Scanscope AT turbo. Immunofluorescence was performed following a standard protocol (Seow et al, 2019). Antibodies and stains used are listed in Reagents

and tools table. Sections were imaged using a Zeiss LSM 980 confocal microscope with a ×63 objective. In total, 25–30 z-slices with an interval of 0.5 µm were imaged to ensure the entire primary cilia were captured. Four images (~100 cilia) per animal were captured, for $n = 3–4$ animals per experimental group. IMCD3 cells grown on coverslips were fixed with 4% paraformaldehyde for 10 min at room temperature and washed three times with DPBS for 3 min each. Fixed cells were permeabilised with 0.1% triton X-100 (Sigma-Aldrich) diluted in DPBS for 10 min and washed three times with DPBS. Permeabilised cells were incubated with block buffer (5% donkey serum in PBST) at room temperature for 30 min. Cells were incubated with primary antibodies (Reagents and tools table) at room temperature for 1 h, then washed three times with DPBS for 5 min each and incubated with secondary antibodies (Reagents and tools table) at room temperature in the dark for 1 h. Cells were washed with DPBS for 5 min each and mounted with ProLong gold antifade moutant (Thermo Fisher Scientific). Cells were imaged with a z-stack using a Leica SP8 confocal microscope with a 63x objective. Overall, 10–15 z-slices with an interval of 0.5 µm were imaged to ensure the entire cilium was captured. Four images (~100 cilia) per biological replicates were taken, where $n$ = three biological replicates per treatment. Confocal immunofluorescence z-stack images were used for primary cilia analysis. Primary cilia were measured manually with the polygon measurement tool in Imaris imaging software (version 9.8.1). Primary cilia per nucleus ratio were counted with FIJI.

## Western blot analysis

Protein was extracted from embryonic kidneys and IMCD3 cells using Radioimmunoprecipitation (RIPA) buffer (150 mM NaCl, 50 mM Tris-HCL pH 8.0, 1% IGEPAL, 0.5% sodium deoxycholate, 0.1% SDS, protease inhibitor and phosphatase inhibitor) and protein lysates (10–20 µg) analysed by western blot as previously described (Short et al, 2020). All antibodies are listed in the reagents and tools table. Membranes were also probed with a β-actin antibody (1:50,000; Sigma-Aldrich A5316) to control protein loading. Blots were incubated with ECL and imaged using X-ray films. The films were scanned and analysed using FIJI imaging software.

## Scanning electron microscopy

Whole kidneys were fixed in 2.5% (v/v) glutaraldehyde and 2% (v/v) paraformaldehyde in 0.1 M sodium cacodylate buffer overnight at 4 °C. Kidneys were washed with three times with 0.1 M sodium cacodylate buffer for 30 min each with rotation at room temperature. Samples were post fixed in 1% (v/v) osmium tetroxide in 0.1 M sodium cacodylate buffer at room temperature for 2 h and washed three times with milliQ water for 30 min each. Following washes, kidneys were bisected into longitudinal halves with scalpel blade. Fixed kidneys were dehydrated in increasing concentrations of ethanol: once in 30%, 50%, 70%, 90% and twice in 100% ethanol for 20 min each. Dehydrated kidneys were dried in an EM CPD300 Critical Point Dryer (Lecia Microsystems) and mounted on 12 mm diameter aluminium scanning electron microscopy stubs using stick carbon tabs. Mounted samples were gold coated with EM Ace600 sputter coater (Lecia Microsystems) and kidneys were imaged with Nova NanoSEM 450 scanning electron microscope (Thermo Fisher Scientific) at a voltage of 2 kV and a spot size of 2.

## Statistical analysis

All statistical analysis was performed using GraphPad Prism statistical analysis software, with statistical significance set at $P < 0.05$ and all error bars as standard error of the mean (SEM). Two groups were compared using two-tailed unpaired $t$ test with unequal variance, and multiple groups were compared by a one-way ANOVA with a Tukey's post hoc test.

# Data availability

RNA-seq datasets reported in this study are available at GEO GSE290962.

The source data of this paper are collected in the following database record: biostudies:S-SCDT-10_1038-S44319-025-00454-0.

# Peer review information

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

## Acknowledgements

This study was funded by an NHMRC Ideas Grant (#1185813) from the National Health & Medical Research Council of Australia and supported in part by the Research Training Program (RTP) Stipend PhD scholarship to JL from Monash University (2022). The authors acknowledge the platform facilities and technical assistance of Monash Animal Research Platforms, Monash Histology Platform, Monash Micro Imaging and Ramaciotti Centre for Cryo-Electron Microscopy at Monash University.

## Author contributions

**Kelly L Short**: Conceptualization; Data curation; Formal analysis; Supervision; Investigation; Methodology; Writing—original draft; Writing—review and editing. **Jianshen Lao**: Resources; Data curation; Formal analysis; Validation; Investigation; Visualization; Methodology; Writing—original draft; Project administration; Writing—review and editing. **Rachel Lam**: Formal analysis; Investigation; Methodology; Writing—original draft; Writing—review and editing. **Julie L M Moreau**: Data curation; Formal analysis; Investigation; Methodology; Writing—original draft; Writing—review and editing. **Judy Ng**: Data curation; Formal analysis; Investigation; Methodology; Writing—original draft. **Mehran Piran**: Data curation; Software; Formal analysis; Validation; Investigation; Methodology. **Alexander N Combes**: Software; Formal analysis; Supervision; Investigation; Writing—original draft; Writing—review and editing. **Denny L Cottle**: Data curation; Formal analysis; Supervision; Validation; Investigation; Methodology; Writing—original draft; Writing—review and editing. **Timothy J Cole**: Conceptualization; Resources; Data curation; Formal analysis; Supervision; Funding acquisition; Validation; Investigation; Methodology; Writing—original draft; Project administration; Writing—review and editing.

Source data underlying figure panels in this paper may have individual authorship assigned. Where available, figure panel/source data authorship is listed in the following database record: biostudies:S-SCDT-10_1038-S44319-025-00454-0.

## Disclosure and competing interests statement

The authors declare no competing interests.

# Expanded View Figures

**Figure EV1.  Single cell analysis and localisation of target gene expression in the fetal mouse kidney at E13.5, E15.5 and E18.5.**

Expression of cell type-specific marker genes within single cell clusters at E13.5 (**A**), E15.5 (**B**) and E18.5 (**C**). Colour scale represents average expression, dot size represents percent of cells within cluster that express the gene.

▶

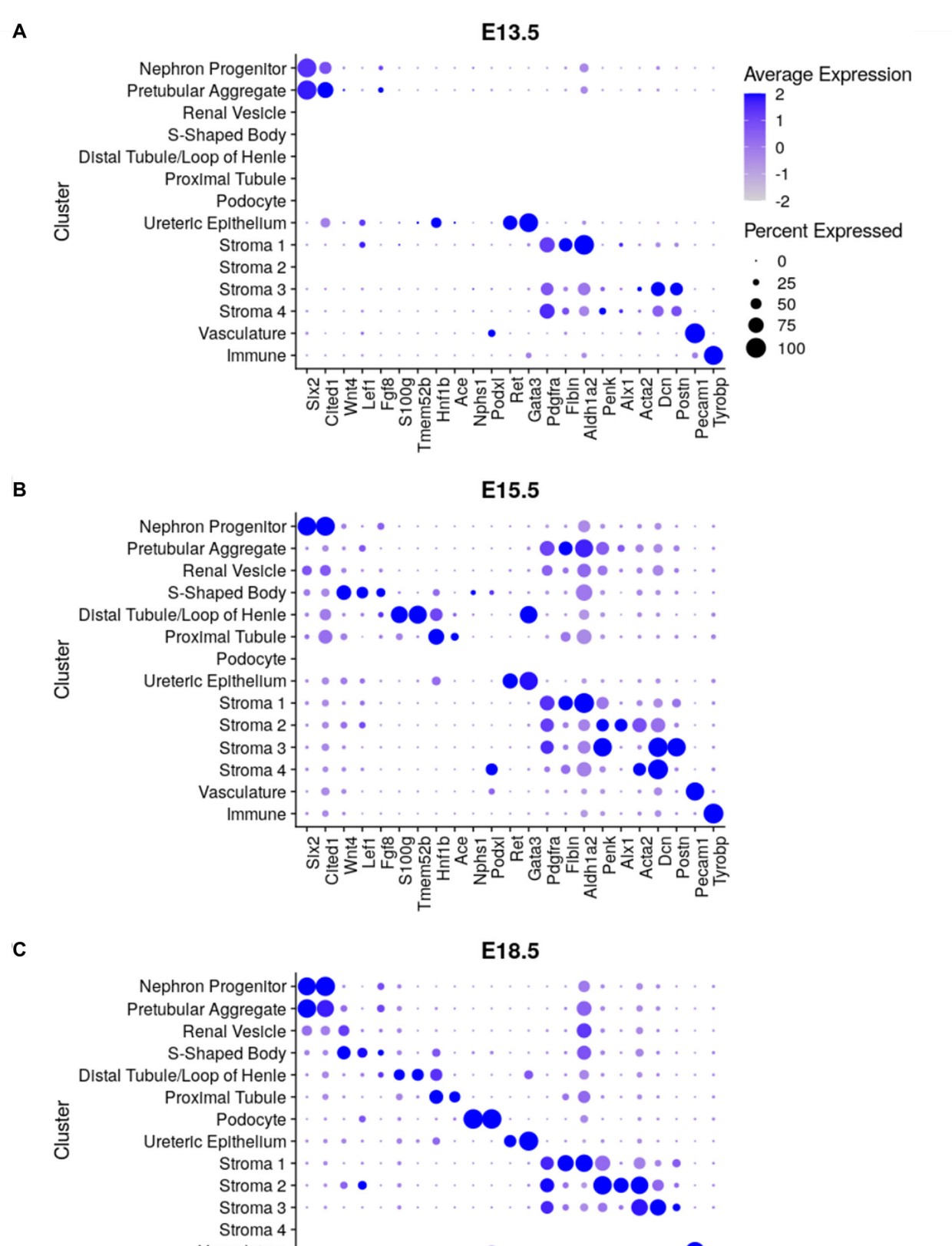

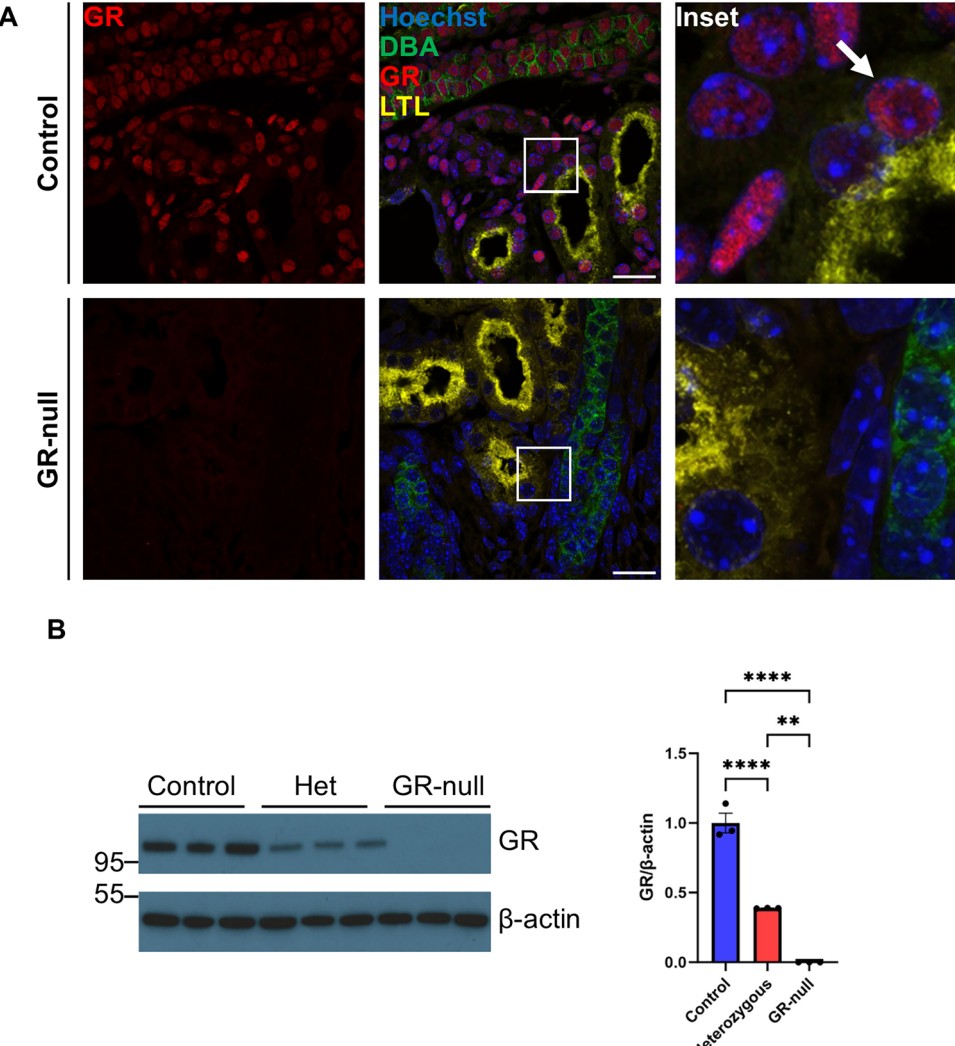

**Figure EV2. Glucocorticoid receptor localisation and deletion in the GR-null fetal mouse kidney.**

(A) Immunofluorescence of glucocorticoid receptor (GR) localisation and deletion in control and GR-null fetal kidney at E18.5. Sections were stained with Hoechst (blue, nucleus), Dolichos Biflorus Agglutinin (DBA) (green, collecting duct), GR (red, GR) and Lotus Tetragonolobus Lectin (LTL) (yellow, proximal tubule). White arrow indicates GR localisation. Slides were imaged with a Zeiss LSM 980 confocal microscopy (63x objective, 2x digital zoom), scale bar represents 20 μm. All images are representative of n = 4 animals per experimental group. (B) Western blot analysis of GR protein in GR-null fetal kidney at E18.5. All data presented as mean ± SEM, significant differences were analysed by one-way ANOVA with multiple comparisons indicated by *$P \le 0.05$, **$P \le 0.01$, ***$P \le 0.001$, ****$P \le 0.0001$, ns=not significant, between control, heterozygous (het) and GR-null. Control vs het ($P = 0.0001$), control vs GR-null ($P = 0.0001$), het vs GR-null ($P = 0.0012$). Data from n = 3 animals per experimental group. Source data are available online for this figure.

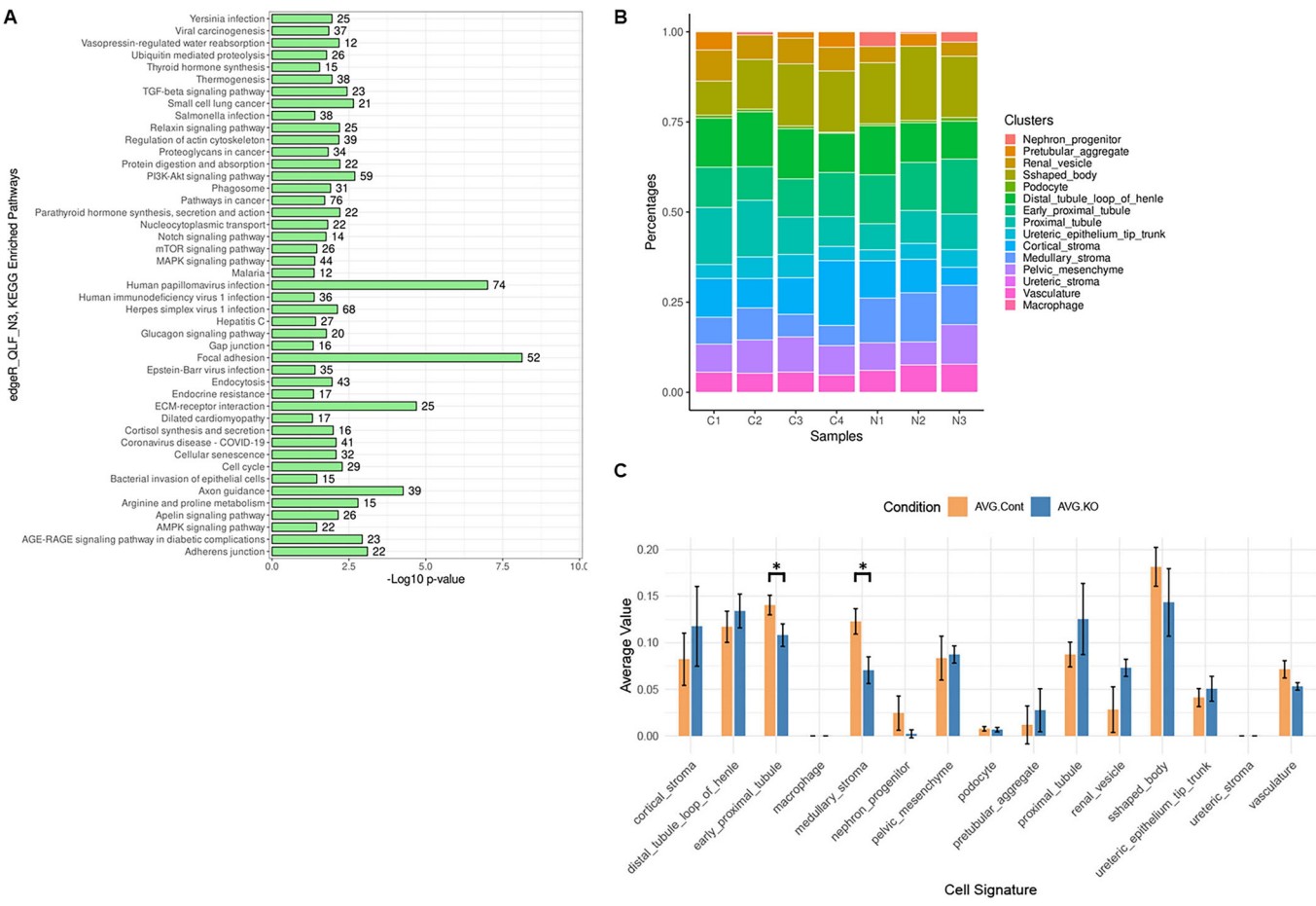

**Figure EV3. Gene set enrichment analysis and RNA-seq deconvolution.**

(A) Gene set enrichment analysis highlights impacted pathways between control and GR-null bulk RNA-seq datasets. RNA-seq was performed on total RNA isolated from control ($n = 4$) and GR-null ($n = 3$) mouse kidneys at E18.5. $P$ value for each term is plotted on the $x$ axis. (B) Bar plot of single cell deconvolution analysis performed to estimate changes in cell signatures in the bulk RNA-seq profiles of control (C1-4) and GR-null (N1-N3) samples. Cell type signatures based on the E18.5 single cell reference are colour coded as per the legend. Analysis of cell proportions between replicates from control and GR-null groups identified small but significant reductions in signatures related to the early proximal tubule ($P = 0.0143$) and medullary stroma ($P = 0.055$) in the GR-null group (two-tailed $t$ test with unequal variance). (C) Bar plot of single cell deconvolution analysis performed to estimate changes in cell signatures in the bulk RNA-seq profiles of control (AVG. Cont, $n = 4$) and GR-null (AVG.KO, $n = 3$) samples. Bars illustrate the percentage average in each group tagged with standard error, significant differences were analysed by two-tailed $t$ test with unequal variance, indicated by *$P \leq 0.05$, **$P \leq 0.01$, ***$P \leq 0.001$, ****$P \leq 0.0001$, ns=not significant. Analysis of cell proportions between replicates from control and GR-null groups identified small but significant reductions in signatures related to the early proximal tubule ($P = 0.0143$) and medullary stroma ($P = 0.055$) in the GR-null group.

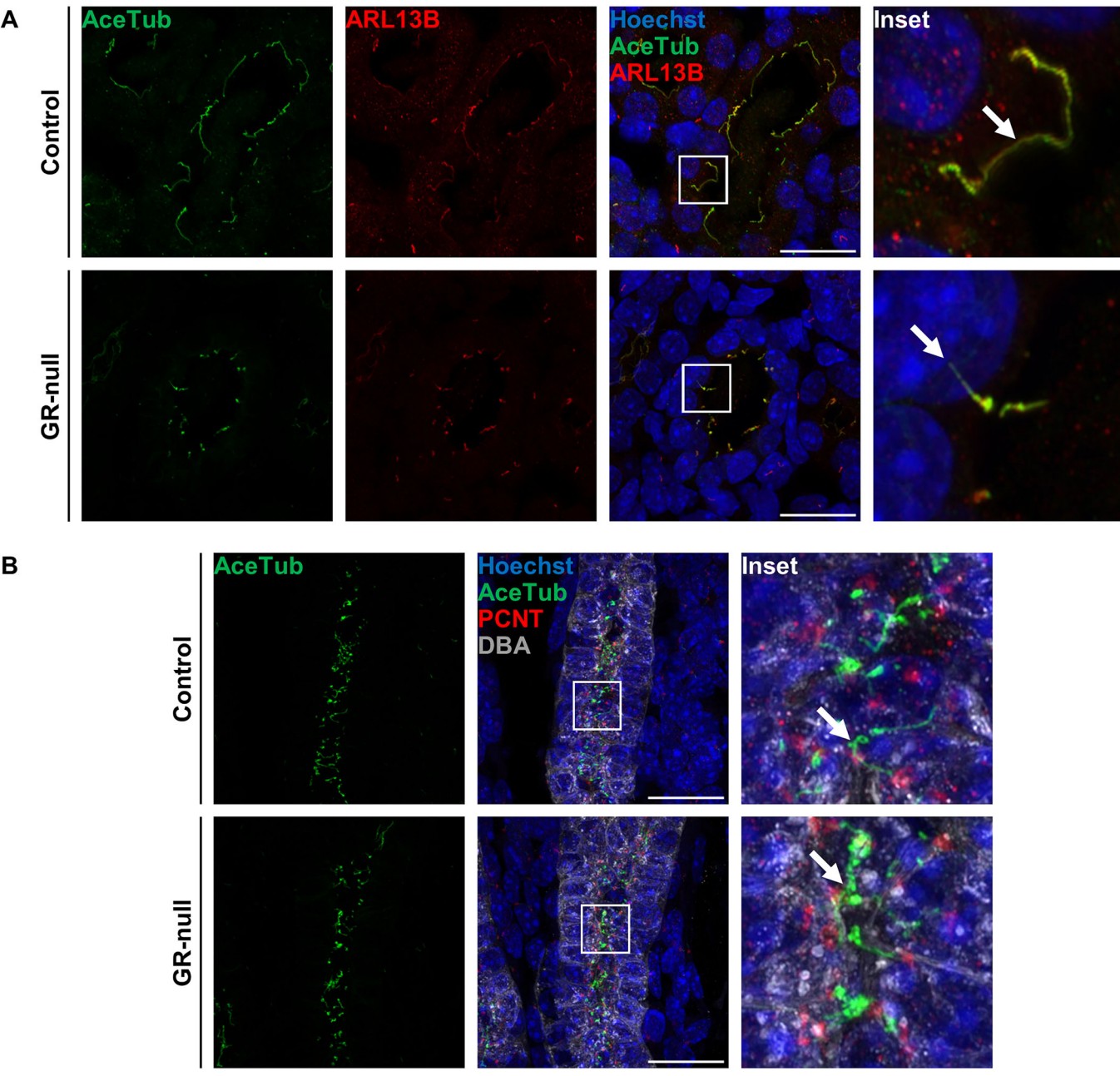

**Figure EV4. Glucocorticoid regulation of primary cilia length in GR-null kidney at E18.5.**

(A) Immunofluorescence of primary cilia morphology in control and GR-null fetal kidney at E18.5. Sections were stained with Hoechst (blue, nucleus), acetylated tubulin (AceTub) (green, microtubules) and ARL13B (red, cilia axoneme). White arrows indicate primary cilia morphology. Slides were imaged with a Zeiss LSM 980 confocal microscopy (×63 objective, 2× digital zoom), scale bar represents 20 μm. All images are representative of $n = 3$ animals per experimental group. (B) Immunofluorescence of primary cilia morphology in control and GR-null collecting ducts at E18.5. Sections were stained with Hoechst (blue, nucleus), acetylated tubulin (AceTub) (green, microtubules), pericentrin (PCNT) (red, basal body) and Dolichos Biflorus Agglutinin (DBA) (grey, collecting duct). White arrows indicate primary cilia morphology. Slides were imaged with a Zeiss LSM 980 confocal microscope (×63 objective, 2× digital zoom), scale bar represents 20 μm. All images are representative of $n = 4$ animals per experimental group. Source data are available online for this figure.

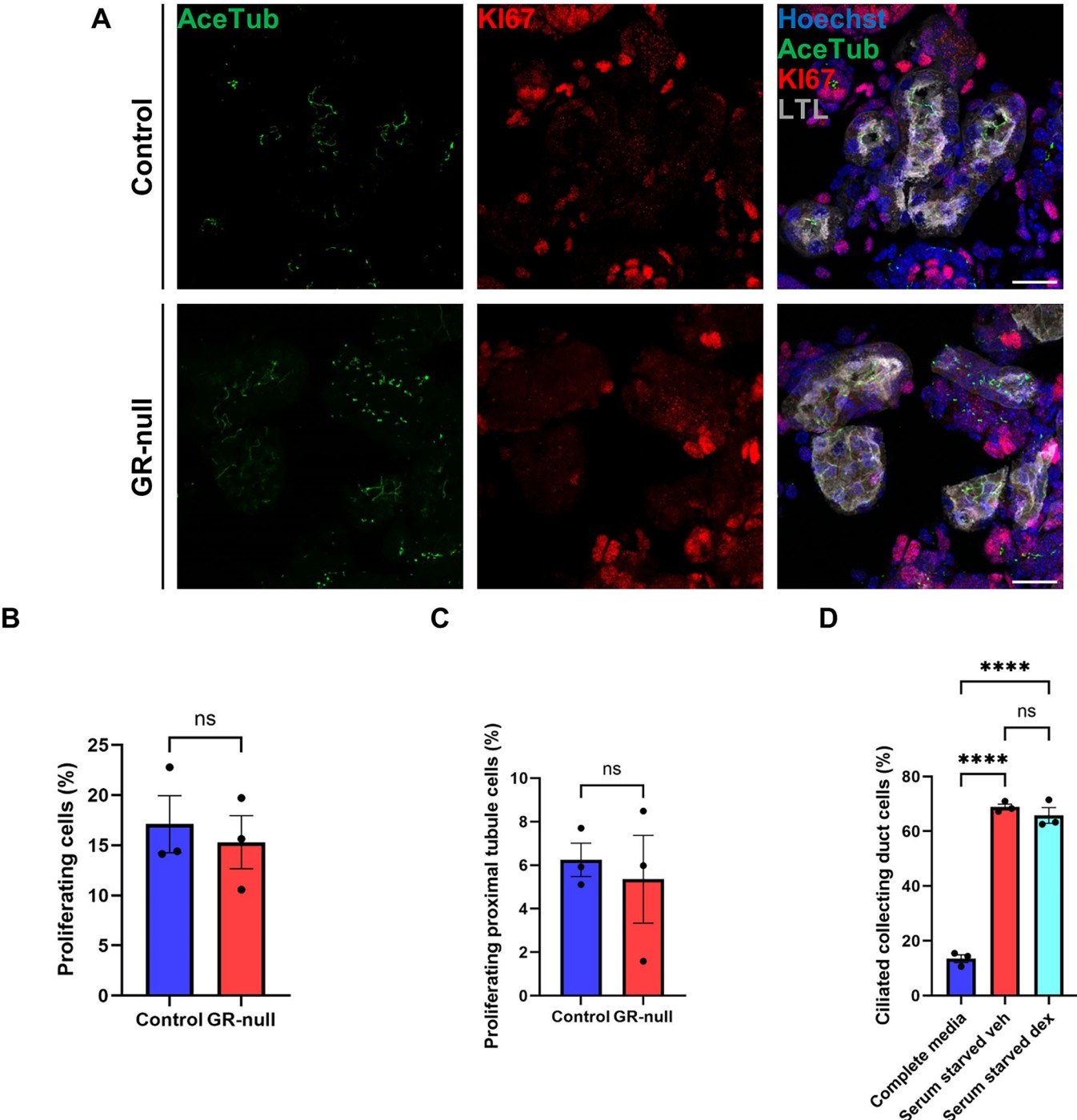

◀ **Figure EV5.  Cell proliferation localisation in the GR-null fetal mouse kidney proximal tubule cells at E18.5 and primary cilia expression in IMCD3 cells after serum starved media conditions.**

(A) Immunofluorescence of proliferation localisation in control and GR-null fetal kidney at E18.5. Sections were stained with Hoechst (blue, nucleus), acetylated tubulin (AceTub) (green, microtubules), KI67 (red, proliferative cells) and Lotus Tetragonolobus Lectin (LTL) (grey, proximal tubule). White arrow indicates proliferative cells. Slides were imaged with a Zeiss LSM 980 confocal microscopy (×63 objective, 2× digital zoom), scale bar represents 20 μm. All images are representative of $n = 4$ animals per experimental group. (B) Percentage of proliferating cells. All data presented as mean ± SEM, significant differences were analysed by unpaired T tests indicated by *$P \le 0.05$, **$P \le 0.01$, ***$P \le 0.001$, ****$P \le 0.0001$, ns=not significant, between control and GR-null ($P = 0.67$), $n = 3$ animals per experimental group. (C) Percentage of proliferating proximal tubule cells. All data presented as mean ± SEM, significant differences were analysed by unpaired T tests indicated by *$P \le 0.05$, **$P \le 0.01$, ***$P \le 0.001$, ****$P \le 0.0001$, ns=not significant, between control and GR-null ($P = 0.70$), $n = 3$ animals per experimental group. (D) Percentage of cells with a primary cilium in complete media versus serum starved vehicle (veh) and serum starved dexamethasone (dex) in IMCD3 cells. All data presented as mean ± SEM, significant differences were analysed by one-way ANOVA with multiple comparisons indicated by *$P \le 0.05$, **$P \le 0.01$, ***$P \le 0.001$, ****$P \le 0.0001$, ns=not significant, between complete media, serum starved veh and serum starved dex. Complete media vs serum starved veh ($P = 0.0001$), complete media vs serum starved dex ($P = 0.0001$), serum starved veh vs serum starved dex ($P = 0.54$). Data from $n = 3$ biological replicates per experimental group. Source data are available online for this figure.

