## [Peer Review File · EMBO Reports]

Disrupted glucocorticoid receptor cell signalling causes a ciliogenesis defect in the fetal mouse renal tubule

Kelly Short, Jianshen Lao, Rachel Lam, Julie Moreau, Judy Ng, Mehran Piran, Alexander Combes, Denny Cottle, and Timothy Cole

Corresponding author(s): Timothy Cole (tim.cole@monash.edu)

Review Timeline:

Submission Date:	24th Jul 24
Editorial Decision:	21st Aug 24
Revision Received:	7th Jan 25
Editorial Decision:	19th Feb 25
Revision Received:	14th Mar 25
Accepted:	4th Apr 25

Editor: Deniz Senyilmaz Tiebe

Transaction Report:

Dear Prof. Cole,

Thank you for transferring your research manuscript to our journal, which was now seen by two referees, whose reports are copied below.

Referees express interest in the proposed role of glucocorticoid receptor cell signalling in regulation of ciliogenesis in the fetal mouse renal tubule. However, they also raise some concerns that need to be addressed to consider publication here. In particular, referee #1 requests more insight into the effect of GR depletion on AurA (point 1). Moreover, referee #2 finds that more controls for DBA stainings, textual clarifications and discussion points need to be added.

I find the reports informed and constructive, and believe that addressing the concerns raised will significantly strengthen the manuscript. Given these positive recommendations, we would like to invite you to submit a revised manuscript. Please revise your manuscript with the understanding that the referee concerns (as in their reports) must be fully addressed and their suggestions taken on board. Please address all referee concerns in a complete point-by-point response. Acceptance of the manuscript will depend on a positive outcome of a second round of review. It is EMBO reports policy to allow a single round of major experimental revision only and acceptance or rejection of the manuscript will therefore depend on the completeness of your responses included in the next, final version of the manuscript.

We realize that it is difficult to revise to a specific deadline. In the interest of protecting the conceptual advance provided by the work, we recommend a revision within 3 months. Please discuss the revision progress ahead of this time with me if you require more time to complete the revisions, or if you have questions or comments regarding the revision (also by video chat).

1. A data availability section providing access to data deposited in public databases is missing (where applicable).
2. Your manuscript contains statistics and error bars based on $n=2$. Please use scatter plots in these cases.

You can submit the revision either as a Scientific Report or as a Research Article. For Scientific Reports, the revised manuscript can contain up to 5 main figures and 5 Expanded View figures, and it should not exceed 27000 characters. If the revision leads to a manuscript with more than 5 main figures it will be published as a Research Article. In this case the Results and Discussion section should be separate. If a Scientific Report is submitted, these sections have to be combined. This will help to shorten the manuscript text by eliminating some redundancy that is inevitable when discussing the same experiments twice. In either case, all materials and methods should be included in the main manuscript file.

4) a .docx formatted letter INCLUDING the reviewers' reports and your detailed point-by-point responses to their comments. As part of the EMBO publication's Transparent Editorial Process, EMBO reports publishes online a Review Process File (RPF) to accompany accepted manuscripts. This File will be published in conjunction with your paper and will include the referee reports, your point-by-point response and all pertinent correspondence relating to the manuscript.

<https://www.embopress.org/page/journal/14693178/authorguide#transparentprocess>

5) a complete author checklist, which you can download from our author guidelines

<https://www.embopress.org/page/journal/14693178/authorguide>. Please insert information in the checklist that is also reflected in the manuscript. The completed author checklist will also be part of the RPF.

6) Please note that all corresponding authors are required to supply an ORCID ID for their name upon submission of a revised manuscript (<<https://orcid.org/>>). Please find instructions on how to link your ORCID ID to your account in our manuscript tracking system in our Author guidelines

<<https://www.embopress.org/page/journal/14693178/authorguide#authorshipguidelines>>

7) Before submitting your revision, primary datasets produced in this study need to be deposited in an appropriate public database (see <https://www.embopress.org/page/journal/14693178/authorguide#datadeposition>). Please remember to provide a reviewer password if the datasets are not yet public. The accession numbers and database should be listed in a formal "Data Availability" section placed after Materials & Method (see also

<https://www.embopress.org/page/journal/14693178/authorguide#datadeposition>). Please note that the Data Availability Section is restricted to new primary data that are part of this study. * Note - All links should resolve to a page where the data can be accessed. *

Additional information on source data and instruction on how to label the files are available:

<https://www.embopress.org/page/journal/14693178/authorguide#sourcedata>

9) Our journal encourages inclusion of *data citations in the reference list* to directly cite datasets that were re-used and obtained from public databases. Data citations in the article text are distinct from normal bibliographical citations and should directly link to the database records from which the data can be accessed. In the main text, data citations are formatted as follows: "Data ref: Smith et al, 2001" or "Data ref: NCBI Sequence Read Archive PRJNA342805, 2017". In the Reference list, data citations must be labeled with "[DATASET]". A data reference must provide the database name, accession number/identifiers and a resolvable link to the landing page from which the data can be accessed at the end of the reference. Further instructions are available at <http://www.embopress.org/page/journal/14693178/authorguide#referencesformat>

10) Regarding data quantification (see Figure Legends:

<https://www.embopress.org/page/journal/14693178/authorguide#figureformat>)

12) Please also note our reference format:

13) All Materials and Methods need to be described in the main text using our 'Structured Methods' format, which is required for all research articles. According to this format, the Methods section includes a Reagents and Tools Table (listing key reagents, experimental models, software and relevant equipment and including their sources and relevant identifiers) followed by a Methods and Protocols section describing the methods using a step-by-step protocol format. The aim is to facilitate adoption of the methodologies across labs. More information on how to adhere to this format as well as a downloadable template (.docx) for the Reagents and Tools Table can be found in our author guidelines:

I look forward to seeing a revised version of your manuscript when it is ready. Please let me know if you have questions or comments regarding the revision.

Kind regards,

Deniz Senyilmaz Tiebe

Deniz Senyilmaz Tiebe, PhD
Senior Scientific Editor
EMBO Reports

Referee #1:

In this manuscript, Short et al. link glucocorticoid signalling to the formation of primary cilia in kidney tubules. They begin by analyzing glucocorticoid receptor (GR) expression in embryonic kidneys. Subsequently, the authors demonstrate that in the embr. kidneys of a glucocorticoid receptor knockout mouse, a significant representation of ciliary proteins is found among a total of 288 deregulated genes. Tissue analysis reveals shortened cilia in the proximal tubule epithelium. Due to perinatal lethality in the conventional knockout model, they specifically delete the GR in the kidney using the HoxB7 Cre line, which also results in shortened cilia in the collecting duct. There is a somewhat tangential discussion on podocytes that does not fully align with the paper's primary focus. In kidney organoids, treatment with dexamethasone leads to elongated cilia in podocyte precursor cells, consistent with the shortened cilia observed in the GR knockout mouse. Dexamethasone also induces cilia elongation in mIMCD3 cells, aligning with the paper's main findings. Inhibition of GR appropriately abolishes this effect. Lastly, the authors note a reduction in AurA expression in the embryonic kidneys of the knockout mouse, potentially providing insight into the mechanisms leading to cilia shortening.

Overall, I find this manuscript highly relevant and extremely interesting. A very well written and designed story. The topic is both exciting and novel, offering numerous potential avenues for translational research. However, I would like to suggest a few points for improvement:

- 1) The role of AurA in this context is not sufficiently explained. Does GR alter AurA transcription, or is the regulation post-translational? Is there a change in the stability or activity of the kinase? Is HEF1 involved? Are there detectable changes in tubulin acetylation?
- 2) The discussion should emphasize that stabilizing cilia may not be an appropriate therapeutic goal for ADPKD. This was demonstrated by studies from Ma et al. (Somlo Lab in ~2013), which suggested that cilia stabilization could exacerbate ADPKD. In the long term, this needs to be tested in a genetic ADPKD model with concurrent GR knockout. This point should be thoroughly discussed, especially considering that ADPKD patients will very likely read this study.

Minor Points:

- 1) What happens to cilia on tubule cells (PT/DT) in the organoid system?
- 2) Adding WT1 or MafB to the Nphs1 stainings would help identify podocytes in both tissue and organoid samples, though this may increase background in embryonic tissues.
- 3) Figure 1 legend: What does "2x zoom" refer to?
- 4) Figure 1H: It would be helpful to show the individual channels.
- 5) If tubulin acetylation is affected by GR knockout, it would be beneficial to confirm the ciliary phenotypes using Arl13B or another cilia marker in IF.

Referee #2:

In this manuscript the authors convincingly show that GR signalling plays a role in determining cilium length in kidney tubule cells. The data presented here provide a firm basis for further analysis of the functional consequences of the regulation of cilium length by GR signalling.

Major comments

I found figure 2 a bit difficult to interpret. It would help if the authors mention in the text (lines 219-220) where the markers that they used and GR localize, e.g. in the cytoplasm, nucleus etc. In addition, based on the DBA staining it was clear to me that GR stained cells of the collecting duct at E14.5, but localization to the stroma was not clear to me. Please use arrows to indicate where this can be seen or explain it in words. Furthermore, the authors state GR is absent in the podocytes at E14.5, but I don't see any podocyte staining.... Moreover, the authors claim GR localization in the mesenchyme, but no marker has been used. Please indicate where this can be seen and/or explain in words in the text. Finally, the authors claim that their staining implicates that GR signalling has important roles in these structures. I think "implicating" is too strong. Please change this to "suggesting".

Line 250 and Sup fig 4b shows small but significant changes in cell populations. Statistics for this has been mentioned in the legend to the figure. From the figure itself it is clear that the different cell populations vary but how they differ between control and GR-null kidneys is quite unclear. If the authors want to make such a claim, they should present a figure that more clearly compares control to GR-null populations. Please include an additional panel that compares the averages, with SD and statistics. I was surprised that although the authors quantify the levels of some proteins or P-proteins in Figure 7, they only include 2 "more structural" cilia proteins (IFT88 and acetylated tubulin) and do not include the proteins encoded by the genes that were differentially expressed in their RNAseq experiment. Why has this not been done? Actually, IFT88 and acetylated tubulin levels were not affected in Fig 7, but the authors do not discuss this. Please include a discussion on this issue.

In line 393, the authors state they could not analyze kidney cyst development because global GR deletion is lethal. However, in this manuscript the authors also use kidney specific GR knock out mice. Do they develop kidney cysts? Please include relevant data.

The authors conclude (e.g. in line 24 and 410) that GR signalling is essential for normal cilia formation. They convincingly show with various experiments that cilium length is affected, but they always see that the number of ciliated cells is not affected. From the data it is not clear if cilium length is affected because of a defect in formation or maintenance of correct cilium length or perhaps increased shedding of cilia. Please discuss this issue.

I don't think that the model, Figure 8 really adds much to the discussion presented in the paper. It can be used as a graphic summary, but does not necessarily have to be included in the paper.

Minor comments

In the abstract, line 25, the authors mention GR-null mice. Please explain what "GR" means.

Several times the authors use "cilium" where "cilia" should have been used (line 27, 176, 185,

Line 39, I think "important key roles" is a bit of an overstatement. Please remove either "important" or "key".

Line 40 states that defects in ciliogenesis cause human ciliopathies. Please change "ciliogenesis" in "cilia", as in many cases the defects affect cilia function and not per se ciliogenesis.

Please introduce a space between a number and the unit, e.g. change 10um to 10 um.

Please remove "sodium and" in line 73.

Please change "signal" into "single" in line 98.

Please change "Differential" into "Differentially" in line 122

There is a mix up of Supplementary Tables 1 and 2, please check.

In line 217 the authors "compare localization of GR", but it is not clear to what they compare GR localization. Please rephrase.

In line 224 the authors refer to supplementary Fig 2. It is not clear why, please explain.

Line 273 and 298, please change "were" to "was".

Line 346, please correct "control".

Line 348, please rephrase "to measure several signalling pathways and ciliary proteins".

Line 404, please correct "protomer".

Subject: EMBOR-2024-60065-T - Response to Referee Reports.**08/01/2025**

Deniz Senyilmaz Tiebe, PhD,
Senior Scientific Editor,
EMBO Reports.

Dear Dr Tiebe,

Please find below a detailed response from the authors to the two referee reports for the manuscript EMBOR-2024-60065. We have responded to each query and request of the referees, and added additional data to the manuscript that we feel strengthens our findings and have updated the manuscript text in line with the referee's suggestions. In particular we have addressed and updated the manuscript for the main concerns of referee #1, providing more discussion on the effect of GR depletion on AurA. With regard to referee #2, as suggested we have added more controls for the DBA staining, and added clarification and additional discussion where suggested by this referee.

We hope you find the revised manuscript acceptable for publication in EMBO Reports.

Yours sincerely,

Prof Timothy J. Cole

Referee #1:

In this manuscript, Short et al. link glucocorticoid signalling to the formation of primary cilia in kidney tubules. They begin by analysing glucocorticoid receptor (GR) expression in embryonic kidneys. Subsequently, the authors demonstrate that in the embr. Kidneys of a glucocorticoid receptor knockout mouse, a significant representation of ciliary proteins is found among a total of 288 deregulated genes. Tissue analysis reveals shortened cilia in the proximal tubule epithelium. Due to perinatal lethality in the conventional knockout model, they specifically delete the GR in the kidney using the HoxB7 Cre line, which also results in shortened cilia in the collecting duct. There is a somewhat tangential discussion on podocytes that does not fully align with the paper's primary focus. In kidney organoids, treatment with dexamethasone leads to elongated cilia in podocyte precursor cells, consistent with the shortened cilia observed in the GR knockout mouse. Dexamethasone also induces cilia elongation in mIMCD3 cells, aligning with the paper's main findings. Inhibition of GR appropriately abolishes this effect. Lastly, the authors note a reduction in AurA expression in the embryonic kidneys of the knockout mouse, potentially providing insight into the mechanisms leading to cilia shortening.

Overall, I find this manuscript highly relevant and extremely interesting. A very well written and designed story. The topic is both exciting and novel, offering numerous potential avenues for translational research. However, I would like to suggest a few points for improvement:

1. The role of AurA in this context is not sufficiently explained. Does GR alter AurA transcription, or is the regulation post-translational? Is there a change in the stability or activity of the kinase? Is HEF1 involved? Are there detectable changes in tubulin acetylation?

Response: *We have measured mRNA levels of *Atat1* (tubulin acetyltransferase 1) and *Nedd9* (*Hef1*) in E18.5 GR-null kidney compared to WT control and detected no significant changes. This is now shown in the revised Figure 2 as Panel G. We have now added more discussion on how GR-signalling may regulate and alter Aurka, potentially via post-translational mechanisms, and future approaches that could be undertaken (Revised manuscript, lines 494-502).*

2. The discussion should emphasize that stabilizing cilia may not be an appropriate therapeutic goal for ADPKD. This was demonstrated by studies from Ma et al. (Somlo Lab in ~2013), which suggested that cilia stabilization could exacerbate ADPKD. In the long term, this needs to be tested in a genetic ADPKD model with concurrent GR knockout. This point should be thoroughly discussed, especially considering that ADPKD patients will very likely read this study.

Response: *The Discussion section has been modified and expanded to include a discussion on cystogenesis and ADPKD where we have referred to and cited the study of Ma et al. (2013). We agree that testing the impact of GR ablation in a genetic model of ADPKD would be an essential future approach to assess the impact of GR-signalling in PKD and have added this point to the discussion section (revised manuscript, lines 461-468).*

Minor Points:

1. What happens to cilia on tubule cells (PT/DT) in the organoid system?

Response: *This is a very good question. Unfortunately, we have been unable to generate additional human kidney organoids due to a lack of initial samples and the associated time constraints but this will be strongly considered in future follow-up studies. We have added a brief discussion on this topic in the revised manuscript (revised manuscript, lines 456-458).*

2. Adding WT1 or MafB to the Nphs1 staining's would help identify podocytes in both tissue and organoid samples, though this may increase background in embryonic tissues.

Response: *This is a good suggestion. As discussed above due to time constraints and sample availability we have been unable to generate additional kidney organoids for study by this will be considered in future experiments.*

3. Figure 1 legend: What does "2x zoom" refer to?

Response: *Apologies for the confusion here. We have changed this in the manuscript to "digital zoom" which refers to enlarging the area visualised (two-fold) without changing the microscope objective.*

4. Figure 1H: It would be helpful to show the individual channels.

Response: *In the revised manuscript the individual immunofluorescence channels for Figure 1H are now presented in Appendix Figures S1 and S2. Figure 1H also contains an additional panel using a mesenchymal stromal marker MEIS123 that shows co-staining with the GR in stromal cells.*

5. If tubulin acetylation is affected by GR knockout, it would be beneficial to confirm the ciliary phenotypes using Arl13B or another cilia marker in IF.

Response: *In Figure 7I of the revised manuscript we show that acetylated tubulin protein levels did not change in the GR-null kidney at E18.5. We have used an additional panel has been added in Appendix Figure S4 which includes an acetylated tubulin and ARL13 immunofluorescence to show co-localisation of microtubules (acetylated tubulin) with ciliary axoneme (ARL13B).*

Referee #2:

In this manuscript the authors convincingly show that GR signalling plays a role in determining cilium length in kidney tubule cells. The data presented here provide a firm basis for further analysis of the functional consequences of the regulation of cilium length by GR signalling.

Major comments

1. I found figure 2 a bit difficult to interpret. It would help if the authors mention in the text (lines 219-220) where the markers that they used and GR localize, e.g. in the cytoplasm, nucleus etc. In addition, based on the DBA staining it was clear to me that GR stained cells of the collecting duct at E14.5, but localization to the stroma was not clear to me. Please use arrows to indicate where this can be seen or explain it in words. Furthermore, the authors state GR is absent in the podocytes at E14.5, but I don't see any podocyte staining.... Moreover, the authors claim GR localization in the mesenchyme, but not marker has been used. Please indicate where this can be seen and/or explain in words in the text. Finally, the authors claim that their staining implicates that GR signalling has important roles in these structures. I think "implicating" is too strong. Please change this to "suggesting".

Response: Our apologies for the confusion to Figure 1H (not Figure 2). We have clarified that podocytes are in fact absent at E14.5 and modified the text accordingly. We have performed immunofluorescent co-localization with the mesenchymal stromal cell marker MEIS123 that shows co-staining with the GR. This data has been added with an additional panel now in the modified Figure 1H. The individual channel fluorescent images are shown in Appendix Figures 1 and 2. As suggested at line 255 of the revised manuscript we changed 'implicating' to 'suggesting'.

2. Line 250 and Sup fig 4b shows small but significant changes in cell populations. Statistics for this has been mentioned in the legend to the figure. From the figure itself it is clear that the different cell populations vary but how they differ between control and GR-null kidneys is quite unclear. If the authors want to make such a claim, they should present a figure that more clearly compares control to GR-null populations. Please include an additional panel that compares the averages, with SD and statistics.

Response: We have clarified the changes in cell populations with an additional panel (as requested) in Supplementary Figure 4c that more clearly compares the cell populations that are significantly changed between WT control and GR-null in the fetal kidney.

3. I was surprised that although the authors quantify the levels of some proteins or P-proteins in Figure 7, they only include 2 "more structural" cilia proteins (IFT88 and acetylated tubulin) and do not include the proteins encoded by the genes that were differentially expressed in their RNAseq experiment. Why has this not been done? Actually, IFT88 and acetylated tubulin levels were not affected in Fig 7, but the authors do not discuss this. Please include a discussion on this issue.

Response: We have now performed additional western blot analysis using available antibodies to measure protein levels of the cilia structural proteins Cep290 and Kif3a. Although we see similar levels in whole kidney extracts we discuss that to see changes in protein levels in renal epithelial cells may require more careful analysis in isolated and purified primary renal epithelial cells. Antibodies to other structural cilia proteins need to be assessed and optimized for western analysis but unfortunately this is beyond the time-frame of the current study. We have added a brief discussion on this issue (lines 422-425). We also have added further discussion on the two structural cilia proteins IFT88 and acetylated tubulin assessed in Fig. 7 and GR-regulated mechanisms (lines 441-447).

4. In line 393, the authors state they could not analyse kidney cyst development because global GR deletion is lethal. However, in this manuscript the authors also use kidney specific GR knock out mice. Do they develop kidney cysts? Please include relevant data.

Response: We did not observe renal cysts in the GRcdKO HoxB7-Cre mice at P11 (see supplementary figure 3 that shows normal kidney histology with no presence of cysts). GR-null HoxB7-Cre mice have also not demonstrated cysts up to 6 months of age. Analysis of further aged mice will be undertaken but remains beyond the time-frame of the current study.

5. The authors conclude (e.g. in line 24 and 410) that GR signalling is essential for normal cilia formation. They convincingly show with various experiments that cilium length is affected, but they always see that the number of ciliated cells is not affected. From the data it is not clear if cilium length is affected because of a defect in formation or maintenance of correct cilium length or perhaps increased shedding of cilia. Please discuss this issue.

Response: We agree with the reviewer that the mechanism behind cilia shortening is not clear and agree it could arise from a defect in formation of cilia length maintenance. We have now included additional data in Figure 3. Panel E that shows scanning electron microscopy images of fetal kidney cells at E18.5. Images from GR-null mice show the presence of abnormally shaped cilia that we suggest contribute to cilia shortening. We have added additional discussion of this data and the possible mechanisms (lines 474-478).

6. I don't think that the model, Figure 8 really adds much to the discussion presented in the paper. It can be used as a graphic summary, but does not necessarily have to be included in the paper.

Response: We agree to this suggestion from the reviewer. We have therefore removed Figure 8 from the main list of figures and will modify this for consideration as a graphic summary or abstract.

Minor comments

In the abstract, line 25, the author mention GR-null mice. Please explain what “GR” means.

Response: Apologies, this stands for ‘glucocorticoid receptor’ and has been added to line 24 in the revised manuscript.

Several times the authors use “cilium” where “cilia” should have been used (line 27, 176, 185).

Response: Revised manuscript, lines 26, 189, 201. This has been corrected to cilia.

Line 39, I think “important key roles” is a bit of an overstatement. Please remove either “important” or “key”.

Response: Revised manuscript, line 38; this has been changed to ‘important roles’.

Line 40 states that defects in ciliogenesis cause human ciliopathies. Please change “ciliogenesis” in “cilia”, as in many cases the defects affect cilia function and not per se ciliogenesis.

Response: Revised manuscript, line 39. This has been changed.

Please introduce a space between a number and the unit, e.g. change 10um to 10 um.

Response: A space has been added where required.

Please remove “sodium and” in line 73.

Response: Revised manuscript, line 77. This has been removed.

Please change “signal” into “single” in line 98.

Response: Revised manuscript, line 106. This has been amended.

Please change “Differential” into “Differentially” in line 122.

Response: Revised manuscript, line 133. This has been amended.

There is a mix up of Supplementary Tables 1 and 2, please check.

Response: Apologies for the mix up. In the revised manuscript Table 1 has been incorporated into the Materials and methods section in the Reagents and tools table, and Table 2 has now become Table EV1 on Page 45.

In line 217 the authors “compare localization of GR”, but it is not clear to what they compare GR localization. Please rephrase.

Response: Revised manuscript, line 247. This has been rephrased to ‘To compare GR localisation at the protein level at different stages of fetal kidney development we performed immunofluorescence....’

In line 224 the authors refer to supplementary Fig 2. It is not clear why, please explain.

Response: Supplementary Figure 2a (now Appendix Fig S1) also shows GR immunostaining in E18.5 the fetal kidney so complements Fig. 1h-k. This has been changed in the revised manuscript, line 254-255 to: ‘(Fig. 1h, Appendix Fig. S1 and Appendix Fig. S2)’.

Line 273 and 298, please change "were" to "was".

Response: Revised manuscript, lines 313 and 338; This has been changed.

Line 346, please correct "control".

Response: Revised manuscript, line 385; corrected to ‘controlled’

Line 348, please rephrase "to measure several signalling pathways and ciliary proteins".

Response: Revised manuscript, line 387; This has been rephrased to ‘to quantify several signaling pathway kinases and ciliary proteins.’

Line 404, please correct "protomer".

Response: Revised manuscript, line 460; corrected to ‘promoter’.

Dear Prof. Cole,

Thank you for submitting your revised manuscript. It has now been seen by both of the original referees.

As you can see, referees find that the study is significantly improved during revision and recommend publication. However, I need you to address the points below before I can accept the manuscript.

- Please address the remaining concerns of referee #1.
 - In line with the remaining concerns of referee #1, during our routine figure checks, we notice potential re-uses of blots in Figure 7 without any explanation in the respective legends of the panels. Please double check (also the source data) and clarify.
 - Funding information should be complete both in the manuscript text and the manuscript tracking system. We note that funding information is currently missing from the manuscript tracking system.
 - Please add a separate Data Availability section to the manuscript, where datasets and computer code that were generated in the reported study should be listed in a structured manner and placed after the Methods section - i.e. the RNA-seq dataset at https://bridges.monash.edu/collections/Transcriptome_analysis_of_fetal_kidney_RNA_from_GR-null_and_control_mice_at_E18_5/7615235
 - Related to the point above, please deposit the RNA-seq analysis in one of the recommended databases here: <https://www.embopress.org/page/journal/14693178/authorguide#datadeposition> and add a link, which directly resolves to the dataset, into the Data Availability section.
 - Please rename the 'Disclosure statement' section as 'Disclosure And Competing Interests Statement'.
 - Please remove the 'Author Contributions' section from the manuscript text.
 - We note that Fig. 2H is currently not called out in the text.
 - We note that this study reanalyzes previously published datasets - e.g. GSE108291 from Combes et al, 2019, which need to be cited in the form of dataset citation in addition to the publication containing the dataset, as shown here: <https://www.embopress.org/page/journal/14693178/authorguide#referencesformat>
- Reference list:
Hörnberg E, Ylitalo EB, Crnalic S, Antti H, Stattin P, Widmark A, Bergh A, Wikström P (2011) Gene Expression Omnibus GSE29650 (<https://www.ncbi.nlm.nih.gov/geo/query/acc.cgi?acc=GSE29650>). [DATASET]

Hörnberg E, Ylitalo EB, Crnalic S, Antti H, Stattin P, Widmark A, Bergh A, Wikström P (2011) Expression of androgen receptor splice variants in prostate cancer bone metastases is associated with castration-resistance and short survival. PLoS One 6: e19059

In-text citation: "...were grouped based on the relative levels of AR-Vs expressed, mainly AR-V7 (Hörnberg et al, 2011; Data ref: Hörnberg et al, 2011)."

- Appendix figures should be compiled in one Appendix PDF, not uploaded separately. Their nomenclature should be Appendix Figure S1 etc. Appendix figure legends should be removed from manuscript file and placed below corresponding figures in the Appendix PDF.
- Please remove the Reagents and Tools table from the manuscript and submit it as a separate file by using the following template: <https://www.embopress.org/page/journal/14693178/authorguide#structuredmethods>
- We note that Table EV1 in the manuscript should rather be included in the Reagents & Tools table. As such, the Table EV2 should be renamed to Table EV1 with the corresponding callouts; the legend for Table EV2 should be removed from the manuscript file and only placed above the table.
- Section order should be corrected: Title page - Abstract & Keywords - Introduction - Results - Discussion - Methods - Data Availability - Acknowledgements - Disclosure and Competing Interests Statement - References - Figure Legends - Table(s) - Expanded View Figure Legends.
- Our production/data editors have asked you to clarify several points in the figure legends - Figure Legends (main + EV):
 - o Please indicate what */ **/ ***/ **** represents in legends as well; if this represents p value(s), please specify the exact p value in the legend(s) of figure(s) 2E-G; 3C, 4B, 5B, F, G; 6B-D; EV2B, EV5 D.
 - o Please note that information related to n is missing in the legends of figures 2C, D.
 - o Please note that the white arrows are not defined in the legend of figure 6A. This needs to be rectified.
- Papers published in EMBO Reports include a 'synopsis' and 'bullet points' to further enhance discoverability. Both are displayed on the html version of the paper and are freely accessible to all readers. The synopsis includes a short standfirst summarizing the study in 1 or 2 sentences (max 35 words) that summarize the paper and are provided by the authors and streamlined by the handling editor. I would therefore ask you to include your synopsis blurb and 3-5 bullet points listing the key experimental findings.
- In addition, please provide an image for the synopsis. This image should provide a rapid overview of the question addressed in the study but still needs to be kept fairly modest since the image size cannot exceed 550 (width) x 300-600 (height) pixels.

Thank you again for giving us to consider your manuscript for EMBO Reports, I look forward to your minor revision.

Kind regards,

Deniz Senyilmaz Tiebe

--

Deniz Senyilmaz Tiebe, PhD
Senior Scientific Editor
EMBO Reports

Referee #1:

The authors have addressed and responded to all the criticisms I raised in the revised version of the manuscript. Therefore, I have no further objections here.

However, during my revision, I noticed that in Figure 7, the same loading controls were used for the newly added blots in Figures J and L. From the source file, it is not entirely clear to me how the actin blot relates to KIF3A. Were the samples run on an independent gel for the actin staining, or are those restainings? Furthermore, it is unclear to me how exactly the densitometry for the CEP290 blot was performed, as the bands are not clearly defined and delineated.

Additionally, in other parts of the figure, actin blots are shown multiple times. One could think of consolidating these blots into a single panel as they pertain to the same experiment.

Referee #2:

I'm happy with the responses of the authors to my comments and support publication of the manuscript.

EMBOR-2024-60065V2 - Response to Referees

Deniz Senyilmaz Tiebe, PhD
Senior Scientific Editor,
EMBO Reports.

Dear Dr Tiebe,

Please find below a response from the authors to the referee reports for the revised manuscript EMBOR-2024-60065V2. We have responded to the query of referee one and have made other suggested changes and adjustments as requested. We have also included a proposed synopsis and uploaded a draft synopsis image.

Yours sincerely,

Prof Timothy J. Cole

Review Comments:

Thank you for submitting your revised manuscript. It has now been seen by both of the original referees. As you can see, referees find that the study is significantly improved during revision and recommend publication. However, I need you to address the points below before I can accept the manuscript. Please address the remaining concerns of referee #1. In line with the remaining concerns of referee #1, during our routine figure checks, we notice potential re-uses of blots in Figure 7 without any explanation in the respective legends of the panels. Please double check (also the source data) and clarify.

Response to Referees:**Referee #1:**

The authors have addressed and responded to all the criticisms I raised in the revised version of the manuscript. Therefore, I have no further objections here.

However, during my revision, I noticed that in Figure 7, the same loading controls were used for the newly added blots in Figures J and L. From the source file, it is not entirely clear to me how the actin blot relates to KIF3A. Were the samples run on an independent gel for the actin staining, or are those restainings? Furthermore, it is unclear to me how exactly the densitometry for the CEP290 blot was performed, as the bands are not clearly defined and delineated. Additionally, in other parts of the figure, actin blots are shown multiple times. One could think of consolidating these blots into a single panel as they pertain to the same experiment.

Response: *Figures J and L were indeed re-stainings or re-probing of the same western blot filter, apologies for not stating this in the figure legend. This has now been rectified in the text. The P-ERK and ERK blots were also re-probed to generate the P-S6 and S6 blots, respectively, so they share an identical beta-actin control blot. The KIF3A western blot was re-probed for CEP290, where again, they share an identical beta-actin blot image. We have included the beta-actin image twice here for ease of comparison between the blot images. This has now been clarified in the Figure 7 legend*

(Page 42; lines 917-920). The densitometry for the CEP290 blot measured only the top band representing full length CEP290 (at 290 kDa), the other smaller bands were not included in analysis.

Referee #2:

I'm happy with the responses of the authors to my comments and support publication of the manuscript.

Further information and formatting requests:

Funding information should be complete both in the manuscript text and the manuscript tracking system. We note that funding information is currently missing from the manuscript tracking system.

Response: *We have now added the funding information to the manuscript tracking system.*

Please add a separate Data Availability section to the manuscript, where datasets and computer code that were generated in the reported study should be listed in a structured manner and placed after the Methods section - i.e. the RNA-seq dataset at:

https://bridges.monash.edu/collections/Transcriptome_analysis_of_fetal_kidney_RNA_from_GR-null_and_control_mice_at_E18_5/7615235

Response: *We have now added a Data Availability section into the manuscript, with the RNA-seq data linked to the GEO database (lines 500-502).*

Related to the point above, please deposit the RNA-seq analysis in one of the recommended databases here:

<https://www.embopress.org/page/journal/14693178/authorguide#datadeposition> and add a link, which directly resolves to the dataset, into the Data Availability section.

Response: *We have now deposited the RNA-seq data into GEO, the resolvable link to the dataset has been included in the Data Availability section of the manuscript (lines 500-502).*

Please rename the 'Disclosure statement' section as 'Disclosure And Competing Interests Statement'.

Response: *This has been renamed (line 511).*

Please remove the 'Author Contributions' section from the manuscript text.

Response: *This has been removed.*

We note that Fig. 2H is currently not called out in the text.

Response: *Figure 2H has been called out or mentioned in the text (lines 120).*

We note that this study reanalyses previously published datasets - e.g. GSE108291 from Combes et al, 2019, which need to be cited in the form of dataset citation in addition to the publication

containing the dataset, as shown here:

<https://www.embopress.org/page/journal/14693178/authorguide#referencesformat>

Reference list:

Hörnberg E, Ylitalo EB, Crnalic S, Antti H, Stattin P, Widmark A, Bergh A, Wikström P (2011) Gene Expression Omnibus GSE29650 (<https://www.ncbi.nlm.nih.gov/geo/query/acc.cgi?acc=GSE29650>). [DATASET]

Hörnberg E, Ylitalo EB, Crnalic S, Antti H, Stattin P, Widmark A, Bergh A, Wikström P (2011) Expression of androgen receptor splice variants in prostate cancer bone metastases is associated with castration-resistance and short survival. PLoS One 6: e19059

In-text citation: "...were grouped based on the relative levels of AR-Vs expressed, mainly AR-V7 (Hörnberg et al, 2011; Data ref: Hörnberg et al, 2011)."

Response: *We have now added the dataset citation in the methods section of the manuscript, in addition to the publication containing the dataset (line 366) and the reference list (lines 561-563).*

Appendix figures should be compiled in one Appendix PDF, not uploaded separately. Their nomenclature should be Appendix Figure S1 etc. Appendix figure legends should be removed from manuscript file and placed below corresponding figures in the Appendix PDF.

Response: *The Appendix Figures have been compiled into one single PDF and uploaded as a complete file with the table of contents, figures and figure legends. The Appendix Figure legends have now been removed from the manuscript.*

Please remove the Reagents and Tools table from the manuscript and submit it as a separate file by using the following template:

<https://www.embopress.org/page/journal/14693178/authorguide#structuredmethods>

Response: *The Reagents and Tools table have been removed from the manuscript and uploaded as a separate file.*

We note that Table EV1 in the manuscript should rather be included in the Reagents & Tools table. As such, the Table EV2 should be renamed to Table EV1 with the corresponding callouts; the legend for Table EV2 should be removed from the manuscript file and only placed above the table.

Response: *Table EV1 has been moved to the Reagents and Tools table. Table EV2 has been renamed to Table EV1 with the corresponding callouts have been renamed (line 113). The table legend has been removed from the manuscript file and placed above the table.*

Section order should be corrected: Title page - Abstract & Keywords - Introduction - Results - Discussion - Methods - Data Availability - Acknowledgements - Disclosure and Competing Interests Statement - References - Figure Legends - Table(s) - Expanded View Figure Legends.

Response: *The order of the manuscript has been corrected. The Methods section heading has been changed to 'Methods' (line 355), and the Translational Statement has been moved from the Abstract page to the end of the Discussion text (lines 349-354).*

Our production/data editors have asked you to clarify several points in the figure legends - Figure Legends (main + EV):

o Please indicate what */ **/ ***/ **** represents in legends as well; if this represents p value(s), please specify the exact p value in the legend(s) of figure(s) 2E-G; 3C, 4B, 5B, F, G; 6B-D; EV2B, EV5 D.

o Please note that information related to n is missing in the legends of figures 2C, D.

o Please note that the white arrows are not defined in the legend of figure 6A. This needs to be rectified.

Response: *The * indicates the level of significance, which has now been included in the figure legend. The p-values have been stated in all figure legends. The n-value has been specified in legend of Figure 2C (line 730-731) and Figure 2D (lines 735-736). White arrows have now been defined in the legend of Figure 6A (line 878).*

Papers published in EMBO Reports include a 'synopsis' and 'bullet points' to further enhance discoverability. Both are displayed on the html version of the paper and are freely accessible to all readers. The synopsis includes a short standfirst summarizing the study in 1 or 2 sentences (max 35 words) that summarize the paper and are provided by the authors and streamlined by the handling editor. I would therefore ask you to include your synopsis blurb and 3-5 bullet points listing the key experimental findings.

Proposed Synopsis:

Glucocorticoids have demonstrated roles in organogenesis. In the fetal kidney deletion of the glucocorticoid receptor reduced primary cilia length and altered cilia structure, whereas the synthetic steroid dexamethasone stimulated cilia growth in renal tubular cells.

- Gene-targeted ablation of glucocorticoid receptor alters the renal transcriptome of the fetal kidney with dysregulated expression of cilia-related genes.
- The loss of the glucocorticoid receptor causes aberrant ciliary morphology, structure and a reduction in primary cilia length in the renal proximal tubule.
- Treatment of renal collecting duct and proximal tubule cell lines with the synthetic glucocorticoid dexamethasone stimulates primary cilia elongation, an effect inhibited by the glucocorticoid receptor antagonist RU486.

In addition, please provide an image for the synopsis. This image should provide a rapid overview of the question addressed in the study but still needs to be kept fairly modest since the image size cannot exceed 550 (width) x 300-600 (height) pixels.

Response: *We have provided an image for the synopsis. This has been uploaded.*

Prof. Timothy Cole
Monash University
Biochemistry & Molecular Biology
Wellington Road
Clayton
Melbourne, Victoria 3800
Australia

Dear Prof. Cole,

Thank you for submitting your revised manuscript. I have now looked at everything and all is fine. Therefore, I am very pleased to accept your manuscript for publication in EMBO Reports.

Congratulations on a nice work!

Kind regards,

Deniz Senyilmaz Tiebe

--

Deniz Senyilmaz Tiebe, PhD
Senior Scientific Editor
EMBO Reports
